

# An instrument for in-situ measurement of total ozone reactivity

Roberto Sommariva[1], Louisa J. Kramer[1], Leigh R. Crilley[1], Mohammed S. Alam[1], and William J. Bloss[1]

[1]School of Geography, Earth and Environmental Sciences, University of Birmingham, Birmingham, UK

**Correspondence:** R. Sommariva (r.c.sommariva@bham.ac.uk)

**Abstract.** We present an instrument for the measurement of total ozone reactivity ($R_{O3}$), i.e. the reciprocal of the chemical lifetime of ozone ($O_3$) in the troposphere. The Total Ozone Reactivity System (TORS) was developed with the objective to study the role of biogenic organic compounds (BVOCs) as chemical sinks of tropospheric ozone. The instrument was extensively characterized and tested in the laboratory using individual compounds and small plants (lemonthyme, *Thymus citriodorus*) in a Teflon bag and proved able to measure reactivities corresponding to $> 4.5 \times 10^{-5}$ s$^{-1}$, corresponding to 20 ppb of $\alpha$-pinene or 150 ppb of isoprene in isolation – larger than typical ambient levels but consistent with levels commonly found in environmental chamber and enclosure experiments. The functionality of TORS was demonstrated in quasi-ambient conditions with a deployment in a horticultural glasshouse containing a range of aromatic plants. The measurements of total ozone reactivity made in the glasshouse showed a clear diurnal pattern, following the emissions of BVOCs, and is consistent with mixing ratios of tens ppb of monoterpenes and several ppb of sesquiterpenes.

## 1 Introduction

Ozone ($O_3$) is a key component of the troposphere: it is known to be damaging for human health and vegetation, to reduce crop yields, and it is an important greenhouse gas (Monks et al., 2015). Ozone is also a primary source of the OH radical, the main atmospheric oxidant, and acts as an oxidant itself (Johnson and Marston, 2008). Because of its importance to tropospheric chemistry, the ozone budget has long been a subject of considerable interest. Ozone is not directly emitted, but is formed in the atmosphere via photolysis of nitrogen dioxide ($NO_2$), followed by reaction of atomic oxygen with molecular oxygen (R1-R2):

$$NO_2 + h\nu \rightarrow O(^3P) + NO \tag{R1}$$

$$O(^3P) + O_2 + M \rightarrow O_3 + M \tag{R2}$$

Ozone is destroyed in the troposphere via a series of processes, both physical (e.g., dry and wet deposition) and chemical (Monks et al., 2015). The latter involve photolysis (R3-R4) and reactions with a range of inorganic molecules and unsaturated volatile organic compounds (R5-R9):

$$O_3 + h\nu \rightarrow O(^3P) + O_2 \tag{R3}$$





$$O_3 + h\nu \rightarrow O(^1D) + O_2 \tag{R4}$$

$$O_3 + NO \rightarrow NO_2 + O_2 \tag{R5}$$

$$O_3 + NO_2 \rightarrow NO_3 + O_2 \tag{R6}$$

$$O_3 + OH \rightarrow HO_2 + O_2 \tag{R7}$$

$$O_3 + HO_2 \rightarrow OH + 2O_2 \tag{R8}$$

$$O_3 + \text{unsaturated VOCs} \rightarrow \text{products} \tag{R9}$$

Ozone photolysis forms atomic oxygen in ground ($O(^3P)$) and excited ($O(^1D)$) state and primarily reacts with the molecular
oxygen in the atmosphere to reform ozone in a null cycle (R2, R10). However, a small fraction ($\sim$10% in the lower troposphere)
of $O(^1D)$ reacts with water vapour to form OH radicals (R11), a key process for the oxidative capacity of the atmosphere:

$$O(^1D) + O_2 + M \rightarrow O_3 + M \tag{R10}$$

$$O(^1D) + H_2O \rightarrow 2OH \tag{R11}$$

The chemical losses of ozone are one of the least understood parts of the tropospheric ozone budget, in particular because of
the many unknowns related to the abundance of volatile organic compounds (VOCs) and their reactivity with ozone (Johnson
and Marston, 2008; Glasius and Goldstein, 2016). The organic compounds that react with ozone contain one or more double
carbon bonds (e.g., alkenes and dialkenes), and many of these species are emitted by plants during their metabolic processes.
These biogenic VOCs (BVOCs) constitute a large fraction of the carbon loading of the atmosphere: estimates suggest that
the total biogenic sources of VOCs can be 8-10 times larger than the total anthropogenic sources (Williams, 2004; Glasius
and Goldstein, 2016). Isoprene is by far the most important BVOC, accounting for $\sim$50% of the total natural emissions of




non-methane hydrocarbons by mass, followed by monoterpenes (15%), methanol (9%), CO (7%) and a range of other organic compounds which include acetone (4%) and sesquiterpenes (3%) (Guenther et al., 2012).

However, due to the limitations of the analytical techniques used to measure VOCs, it is very likely that both the number and the mass of BVOCs in the troposphere are severely underestimated (Di Carlo et al., 2004; Sinha et al., 2010; Whalley

et al., 2011). Estimates vary depending on the approach used, but it is thought that between 20% and 60% (and possibly more) of the total organic carbon pool in the troposphere is currently unaccounted for (Lewis et al., 2000; Goldstein and Galbally, 2007; Glasius and Goldstein, 2016). A significant fraction of these unmeasured organic compounds is constituted by biogenic VOCs: besides isoprene, only a few monoterpenes (e.g., $\alpha$-pinene, $\beta$-pinene, limonene) and even fewer sesquiterpenes (e.g., $\beta$-caryophyllene) are routinely measured in the atmosphere (Bouvier-Brown et al., 2009; Hellén et al., 2018). Sesquiterpenes

are particularly challenging to measure, due to their low vapour pressure, and therefore their ambient levels are most likely underestimated (Pollmann et al., 2005; Duhl et al., 2008; Kim et al., 2009).

All BVOCs are very reactive with both OH and $O_3$ and, therefore, the existence of a significant pool of unknown and unmeasured BVOCs has important consequences for the quantification of the ozone budget and ambient concentrations, which are crucial to understand the environmental and societal impacts of ozone pollution. The oxidation of BVOCs by ozone is

especially important because it forms additional pollutants, such as secondary organic aerosol, and impacts key chemical processes such as the conversion of NO to $NO_2$ and, therefore, the formation of ozone via R1-R2, as well as the radical budget (Lewis et al., 2000; Glasius and Goldstein, 2016). Missing a large fraction of BVOCs means that all these processes remain potentially underestimated.

One way to address this problem is to expand the number of VOCs that can be measured. This approach has achieved some

success, thanks to comprehensive 2-dimensional gas chromatographic techniques (Pankow et al., 2012; Edwards et al., 2013; Glasius and Goldstein, 2016). However, the chemical complexity of the composition of ambient air makes it difficult, if not impossible, to completely quantify the VOC loading of the atmosphere. An alternative approach is to measure an integrated chemical property of all VOCs, such as the chemical lifetime, which includes all the reactions that remove a given species, in this case $O_3$ (R3-R9). Instruments to measure directly the total ozone reactivity have been demonstrated by Park et al. (2013)

and Matsumoto (2014), and used for laboratory studies of gas-phase ozonolysis reactions (Matsumoto, 2016).

This paper presents an instrument designed to measure total ozone reactivity under ambient, environmental chamber and branch enclosure conditions. The Total Ozone Reactivity System (TORS) was developed from the instrument described by Matsumoto (2014) with several modifications, as described below. Section 2 describes the theoretical and operating principles of TORS, while Sect. 3 and Sect. 4 describe the design and the laboratory characterization of the TORS instrument, respec-

tively. In Sect. 5 the TORS instrument is evaluated with three types of experiments, increasingly approaching ambient conditions: laboratory measurements with known BVOC mixtures, laboratory measurements with small plants, and quasi-ambient measurements in a horticultural glasshouse.





## 2 Ozone Reactivity

### 2.1 Reactivity measurements

The atmospheric lifetime ($\tau$) of a generic species $A$ is defined as the inverse of its total chemical loss rate, i.e. of its chemical reactivity ($R$):

$$\tau_{(A)} = \frac{1}{R_{(A)}} = \frac{1}{\Sigma_i k_i [X]_i} \tag{1}$$

where $[X]_i$ is the concentration of a molecule reacting with $A$ and $k_i$ is the corresponding pseudo first-order rate coefficient. The chemical lifetime of a species has long been considered a useful diagnostic parameter, especially to investigate the loss

terms of key atmospheric oxidants, such as the OH radical, which are generally less well known than the respective production terms (Bell et al., 2003).

The comparison between the lifetime measured directly and the lifetime calculated from the independent determination of $k_i$ and $[X]_i$ (Eq. 1) allows us to understand whether all the loss terms for a given species have been accounted for. This approach has been used with success to investigate the budget of the OH radical (Kovacs and Brune, 2001; Ingham et al., 2009; Sinha

et al., 2010; Whalley et al., 2016; Fuchs et al., 2017; Sanchez et al., 2018) and, more recently, of the NO$_3$ radical (Sobanski et al., 2016; Liebmann et al., 2018). Measurements of OH reactivity have also helped to discover previously unknown chemistry in terms of recycling and losses of the OH radical (Di Carlo et al., 2004; Lou et al., 2010; Whalley et al., 2011).

The same principle can be used to investigate the chemical loss of ozone. From reactions R1-R11, the rate of production/loss of tropospheric ozone can be calculated as:

$$\frac{d[O_3]}{dt} = -k_{NO}[NO] - k_{NO2}[NO_2] - k_{OH}[OH] - k_{HO2}[HO_2] - \Sigma_i k_i[VOC]_i - \alpha j_{O3}[O_3] + j_{NO2}[NO_2] \tag{2}$$

where $\alpha$ is the fraction of atomic oxygen that does not reform O$_3$ via reaction with O$_2$ (R11). Halogens (X = Cl, Br, I) can also react with ozone to form halogen oxides (XO), although this process is likely minor in continental environments far from the main halogen sources (Monks et al., 2015; Simpson et al., 2015). Under the unpolluted conditions of a forested environment (NO = 50 ppt, NO$_2$ = 500 ppt, OH = $5 \times 10^6$ molecule cm$^{-3}$, HO$_2$ = $1 \times 10^8$ molecule cm$^{-3}$, Griffith et al. (2013)) the loss

of ozone to NO$_2$, OH and HO$_2$ is a factor of 50-100 times slower than the loss of ozone to NO (Table 1). Under more polluted conditions, NO$_2$, OH and HO$_2$ reactions are even less important as ozone sinks compared to NO.

During the night – when photolysis is zero, and the concentrations of OH and HO$_2$ are typically 2 orders of magnitude lower than during the day – NO is titrated to NO$_2$ (R5) soon after sunset, leading to the formation of NO$_3$ radicals (R6). The rate coefficient of O$_3$ + NO$_2$ is small ($k = 3.52 \times 10^{-17}$ cm$^3$ molecule$^{-1}$ s$^{-1}$, Table 1) compared to the rate coefficients of

most monoterpenes and sesquiterpenes with O$_3$ (since its emissions are light dependent, isoprene is normally not present at night (Guenther et al., 2012)): for example, 1 ppb of NO$_2$ has the same O$_3$ reactivity as 2.7 ppb of $\alpha$-pinene and 6.2 ppb of limonene. This means that NO$_2$ can be a significant ozone loss during the night only when its concentration is high compared





to the total BVOC loading, which is usually not the case in unpolluted forested environments. Equation 2 can thus be simplified to:

$$\frac{d[O_3]}{dt} = -k_{NO}[NO] - k_{NO2}[NO_2] - \Sigma_i k_i[VOC]_i - \alpha j_{O3}[O_3] + j_{NO2}[NO_2] \tag{3}$$

where the NO and photolysis terms are significant only during the day and the $NO_2$ term is significant only during the night (and under relatively high $NO_x$ conditions). The concentrations of $O_3$, NO and $NO_2$, their rate coefficients with $O_3$ and the photolysis rates of $O_3$ and $NO_2$ can be measured and/or are well known. Therefore, Eq. 3 can be used to evaluate the contribution of the volatile organic compounds ($\Sigma_i k_i[VOC]_i$). As mentioned in Sect. 1, only a handful of the VOCs that react

with ozone are routinely measured, which likely leads to underestimation of the VOC contribution to ozone loss. If the loss rate of ozone can be directly measured, it is possible to determine the total VOC loading and compare it with the measured VOCs, thus allowing an estimate of the missing (i.e., non measured) VOCs.

## 2.2 TORS concept

The Total Ozone Reactivity System (TORS) is based on the technique developed by Matsumoto (2014). At its core, the system

is a dark flow tube reactor in which a known amount of ozone is reacted with a sample (e.g., NO, unsaturated VOCs). If the change in the concentration of the co-reactants in the sample is small, the reaction follows pseudo first-order kinetics and the change in ozone concentration is described by:

$$[O_3]_t = [O_3]_0 \times e^{-k't} \tag{4}$$

where $[O_3]_0$ and $[O_3]_t$ are, respectively, the initial and final ozone concentrations, $k'$ is the pseudo first-order rate coefficient

and $t$ is the reaction time, which corresponds to the reactor residence time and can be determined experimentally (Sect. 4.1). Equation 4 can be solved analytically provided the reaction time and the initial and final $O_3$ concentrations are known:

$$k' = \frac{-ln([O_3]_t/[O_3]_0)}{t} \tag{5}$$

TORS provides a direct measurement of $k'$, which includes the chemical reactions inside the reactor (described by Eq. 3) plus any other $O_3$ removal processes, such as the loss of ozone on the reactor wall. Since the reactor is completely dark, the

photolysis terms in Eq. 3 can be neglected and the only contributors to the $O_3$ chemical loss are NO and VOCs. The focus of this study is on the reactivity of VOCs (Sect. 1), and therefore the loss of $O_3$ due to NO and to the reactor wall need to be subtracted. For the purpose of this work, we define ozone reactivity ($R_{O3}$) as:

$$R_{O3} = k' - R_{NO} - R_{wall} = k' - k_{NO}[NO] - R_{wall} = \Sigma_i k_i[VOC]_i \tag{6}$$





where $R_{\mathrm{NO}}$ is the removal of $O_3$ by reaction with NO and $R_{\mathrm{wall}}$ is the loss of $O_3$ on the reactor wall. $R_{\mathrm{wall}}$ is a potentially

significant parameter in the TORS technique and requires accurate and precise determination (Sect. 4.2). Another potentially important factor is the effect of secondary reactions, which can increase the loss of ozone (R5-R8) causing overestimation of $R_{\mathrm{O3}}$ or can decrease the concentration of VOCs in the sample via reactions other than ozonolysis (e.g., if there is significant formation of $NO_3$ in the reactor via R6) causing underestimation of $R_{\mathrm{O3}}$.

Particularly important can be the formation of OH radicals from VOC ozonolysis (Paulson et al., 1999; Rickard et al., 1999;

Johnson and Marston, 2008; Alam et al., 2011): to remove this interference an OH scrubber, such as cyclohexane, can be added to the system. Cyclohexane does not react with $O_3$, but reacts quickly with OH ($k = 6.97 \times 10^{-12} \ \mathrm{cm^3 \ molecule^{-1} \ s^{-1}}$) and forms organic peroxy radicals which combine with $HO_2$ to form products that do not react with $O_3$ (Alam et al., 2011). Therefore, cyclohexane acts as an efficient OH scrubber when present at high mixing ratios ($\mathrm{ppm}$ level). The effects of ozone loss on the reactor wall and of secondary chemistry of ozonolysis products were investigated with a box-model simulation of

the TORS chemistry (Sect. 2.3) and with laboratory characterization experiments (Sect. 4.2).

## 2.3   Simulation of TORS chemistry

A box-model was used to simulate the chemical reactions occurring in the reactor. The main objective of the model was to assess the effect of the OH radicals generated by the ozonolysis of VOCs and the role of an OH scrubber, as well as the impact of potential interfering chemistry such as removal of ozone by OH, $HO_2$ and $NO_2$ (R6-R8), and removal of VOCs by OH and

$NO_3$.

The model was assembled using the AtChem2 modelling toolkit (Sommariva et al., 2019) with the inorganic chemistry and the oxidation mechanisms of $\alpha$-pinene and cyclohexane from the Master Chemical Mechanism (MCM v3.3.1, Saunders et al. (2003)). The cyclohexane mechanism was updated to include a more accurate representation of the ring-opening path of the cyclohexoxy radical, following Alam et al. (2011), although the model results were not subtantially different from those of a

model using the MCM standard cyclohexoxy radical scheme. It must be noted that the model results depend on the VOC used, as the OH yield from ozone + alkene reactions can vary from 0.16 for ethene to 0.90 for 2,3-dimethyl-2-butene (Johnson and Marston, 2008). The choice of $\alpha$-pinene for the model, as well as for the laboratory experiments (Sect. 5.1) is due to the fact that $\alpha$-pinene is one of the most abundant BVOC (Guenther et al., 2012). It also has a high OH yield (0.8, Johnson and Marston (2008)), allowing an estimate of the upper bounds of the the potential interferences caused by OH chemistry.

The model was initialized with a range of $\alpha$-pinene mixing ratios (0.1-50 $\mathrm{ppb}$) and, for each of them, it was run with a range of cyclohexane mixing ratios (0-20 $\mathrm{ppm}$). The initial mixing ratios of NO and $NO_2$ were 50 and 500 $\mathrm{ppt}$, respectively, representative of an unpolluted forested environment which would be the typical location for TORS measurements (Griffith et al., 2013). The initial $O_3$ mixing ratio was set to 120 $\mathrm{ppb}$, as used during the experimental work (Sect. 4.3). The model runtime was 300 seconds, covering the range of potential instrument residence times (Sect. 4.1). A summary of the model

results is shown in Fig. 1.

The removal of $\alpha$-pinene during the residence time in the reactor was 3-5% and $\sim$7% for initial $\alpha$-pinene mixing ratios of $\geq$0.5 $\mathrm{ppb}$ and 0.1 $\mathrm{ppb}$, respectively (Fig. 1a). In the absence of cyclohexane, the removal of $\alpha$-pinene was slightly higher





(1-2 percentage points) due to reaction with OH radicals. These numbers indicate that the consumption of $\alpha$-pinene inside the reactor was over an order of magnitude smaller than its initial concentration and, therefore, the chemical system approached

the pseudo first-order conditions required for the TORS method to work (Sect. 2.2). The error in the determination of ozone reactivity caused by the assumption of pseudo first-order conditions can be estimated at $<4\%$ for $\alpha$-pinene mixing ratios $>10$ ppb. As expected (Alam et al., 2011), the reactions with oxidation products of cyclohexane were not significant sinks for ozone: apart from the wall loss, which is not included in the model, the loss of $O_3$ inside the reactor was less than 1.5% for $\alpha$-pinene mixing ratios of up to 50 ppb and indipendent on the concentration of cyclohexane (Fig. 1b).

The model results also indicate that the formation of OH radicals from the ozonolysis of $\alpha$-pinene was partially offset by their destruction via reaction with $\alpha$-pinene itself and with its reaction products. Fig. 1c shows the difference between the initial and the final concentrations of the OH radical, as calculated by the model. In the absence of cyclohexane, the model calculated net OH production (up to $3 \times 10^5$ molecule cm$^{-3}$) for $\alpha$-pinene mixing ratios of $<1$ ppb and net OH destruction (up to $1.3 \times 10^5$ molecule cm$^{-3}$) for $\alpha$-pinene mixing ratios of 5 ppb or more (Fig. 1c). Compared to the case with zero

cyclohexane, the concentration of OH in the reactor decreased by 2 orders of magnitude at levels of cyclohexane between 1 and 5 ppm, depending on the $\alpha$-pinene level: increasing the cyclohexane mixing ratio above 5 ppm did not cause further decrease in the calculated concentration of OH, nor a reduction in the loss of ozone and $\alpha$-pinene (Fig. 1a-c).

Figure 1d shows that the ozone reactivities determined with 1 and 20 ppm of cyclohexane were essentially the same for $\alpha$-pinene initial mixing ratios up to 50 ppb. Moreover, the model results show that the differences between the ozone reactivity

calculated with and without cyclohexane were between +1% and -5%, depending on the $\alpha$-pinene level (Fig. 1d). This demonstrates that OH chemistry has a small overall impact on the determination of total ozone reactivity, a conclusion that is supported by the laboratory experiments (Sect. 5.1). Model calculated $HO_2$ concentrations were less than $1 \times 10^8$ molecule cm$^{-3}$, meaning that ozone reactivity with $HO_2$ was two orders of magnitude lower than ozone reactivity with $\alpha$-pinene (at 10 ppb of $\alpha$-pinene). Only at very low concentrations of $\alpha$-pinene ($<0.1$ ppb), $HO_2$ was a significant sink for ozone.

The presence of $NO_3$ radicals in the reactor is a potentially important interference for the TORS technique, both because its formation consumes $O_3$ (R6) and because $NO_3$ reacts quickly with $\alpha$-pinene ($k = 6.2 \times 10^{-12}$ cm$^3$ molecule$^{-1}$ s$^{-1}$). Ambient $NO_3$ is likely lost in the inlet before the reactor, since the transmission of $NO_3$ through the inlet – a 6 mm diameter, 5 m long teflon tube with a residence time of $\sim 4$ seconds – is poor (Dubé et al., 2006). However, $NO_3$ can be formed inside the reactor via R6 and the model calculated $NO_3$ formation of the order of $10^6$ molecule cm$^{-3}$ for $\alpha$-pinene mixing ratios $>5$ ppb.

Although the rate coefficient of $\alpha$-pinene + $NO_3$ is 5 orders of magnitude larger than the rate coefficient of $\alpha$-pinene + $O_3$, the ozone concentration in the reactor is 6-7 orders of magnitude higher than the concentration of $NO_3$. Therefore the reactivity of $\alpha$-pinene with $O_3$ was 1-2 orders of magnitude larger than its reactivity with $NO_3$ (Table 1). It must also be noted that $NO_3$ formation is only an issue for ambient measurements, not for laboratory, enclosure and environmental chamber experiments under low or zero $NO_x$ conditions.

To summarize, the model of the TORS reactor suggest that, under typical operating conditions, the concentrations of $HO_2$ and $NO_3$ are too small to compete with BVOCs for reaction with $O_3$. Additionally, the model provides no indication that the products of the oxidation of cyclohexane, when used as OH scrubber, can significantly affect the determination of total ozone



reactivity. While ppm levels of cyclohexane effectively eliminate the OH radicals formed by BVOC ozonolysis reactions, the model suggests that ozone reactivities determined with and without an OH scrubber differ by <5%. The model results are in agreement with the discussion in Sect. 2.3, where it was concluded that the decay of $O_3$ in the TORS reactor is predominantly due to the reactions with NO and VOCs, and potentially to loss on the reactor wall (Eq. 6). It is important to note, however, that the conclusions of the model simulations may vary depending on the chemical conditions in the reactor. Several factors affect the chemistry inside the TORS reactors: the type and mixture of VOCs in the sample, their OH yields, the ambient concentrations of NO and $NO_2$ and, to a lesser extent, ambient temperature and pressure.

## 3  Instrumentation

### 3.1  Description of TORS

The operating principles of TORS are described in Sect. 2.2 and a diagram of the TORS instrument is shown in Fig. 2. The reactor is a 1 m long polytetrafluoroethylene (PTFE) tube with an external diameter of 90 mm and an internal diameter of 87.33 mm. Several different materials and geometries for the reactor were tested during the instrument development phase (Sect. 4.2) and this design was found to allow a residence time inside the reactor long enough for the ozonolysis reactions to take place to a suitably measurable extent, while minimizing the consumption of VOCs – in order to maintain pseudo-first order conditions – and the loss of ozone on the reactor wall (Sect. 4.1).

An ozone flow is generated by irradiating a flow of zero air with a UV mercury lamp (UVP Ltd., UK). The ozone flow is mixed with the sample flow just before the reactor and the initial ozone concentration ($[O_3]_0$) is measured at this point, while the final ozone concentration ($[O_3]_t$) is measured at the exit point of the reactor (Fig. 2). Depending on the instrument settings, this setup produces an $O_3$ mixing ratio in the reactor of 100-140 ppb. Ozone concentrations are measured using two identical UV photometric $O_3$ monitors (Model 49i, Thermo Fisher Scientific, USA). The Model 49i $O_3$ monitor has a stated detection limit of 1 ppb and a precision of 0.25 ppb at 1 minute averaging time. The reactor can be bypassed using two 3-way teflon valves, so that the two ozone monitors can simultaneously measure the $O_3$ concentration before it enters the the reactor, thus allowing the ozone measurements to be corrected for any difference between the two monitors (Sect. 4.3). In addition, a T/RH probe (HMP110, Vaisala Oyj, Finland) is inserted in the reactor to monitor temperature and relative humidity. All the flows in the TORS instrument are controlled with mass flow controllers (Brooks Instrument LLC, USA) using a custom-built control box (IGI Systems Ltd, UK). The signals from the ozone monitors, the T/RH probe and the mass flow controllers are logged on a laptop and processed with ad-hoc software in the R programming language.

A potentially important factor for TORS is the stability of the ozone source: highly variable levels of $O_3$ in the ozone flow (Fig. 2) can affect the determination of the ozone reactivity and increase the signal-to-noise ratio of the instrument. The ozone mixing ratio generated by the mercury lamp was found to vary, on average, by 0.4-0.6 ppb (5 minutes, 2-$\sigma$), i.e. less than 1%. The ozone reactivity measurements reported in this paper (Sect. 5) were all averaged to 5 minutes.



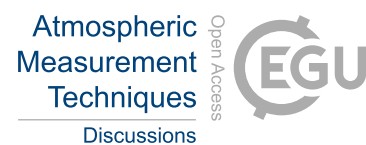

## 3.2 Supporting VOC measurements

A proton-transfer-reaction time-of-flight mass spectrometer (PTR-QiTOF-MS, Ionicon Analytik GmBH, Austria) was used to measure VOC concentrations during the laboratory experiments (Sect. 5.1, Sect. 5.2). The instrument (Sulzer et al., 2014) was operated according to standard operating conditions recommended by the manufacturer (drift pressure = 3.8 mbar, drift tube temperature = 80 °C and E/N = 129 Td), using $H_3O^+$ as the reagent ion. Calibration was performed using a TO-14A aromatics standard mixture (Airgas Inc., USA). This mixture does not contain biogenic compounds, so a mass transmission curve was

used for quantification. Recent work has shown that this approach can be used to quantify uncalibrated species for PTR-MS instruments operating within standard conditions (Holzinger et al., 2019).

  Measurements of BVOCs can be problematic for monoterpenes and sesqiterpenes, as the PTR technique cannot distinguish between isomers (de Gouw and Warneke, 2007). Typically, when measured by PTR-MS the main fragment ions for monoterpenes and sesquiterpenes are at m/z 81.07 and 149.1, respectively, and this fragmentation is independent of the structure of

the isomers (Tani et al., 2003; Kim et al., 2009). Therefore total monoterpene and sesquiterpene concentrations were estimated using the abundances of the protonated parent ions (m/z 137 and 205, respectively) and of the main fragment ions (m/z 81 and 149, respectively). Other compounds associated with biogenic emissions (e.g., substituted monoterpene alcohols, after loss of a neutral $H_2O$) can also be detected at these fragment ions (Kim et al., 2010) and thus the estimated concentrations shown here may be considered upper limits.

The PTR-MS instrument was not available for the experiments outside the laboratory (Sect. 5.3). Instead, samples were taken with adsorption tubes, which were desorbed using a TD Unity-2 thermal desorption unit (Markes Int., UK) and subsequently analysed with a gas chromatograph (GC 7890B, Agilent Technologies, USA) interfaced with a BenchTOF-Select time-of-flight mass spectrometer with tandem ionisation (Markes Int., UK). Further information on the analytical technique and the GC protocol can be found in Alam et al. (2016). The GC-MS analysis were only qualitative due to unavailability of appropriate

calibration standards, with the data used to identify the VOCs in the samples.

## 4 Characterization of TORS

### 4.1 Residence time

Residence time is one of the key parameters in Eq. 5 and therefore needs to be determined as accurately as possible. The geometry of the reactor and the total flow (sample + ozone) determine the residence time. Three methods were used to determine

the residence time: 1) direct measurement using a double injection of acetone 2) indirect measurement via determination of NO reactivity with $O_3$ and 3) calculation using the flow rate and the internal volume of reactor (5990 $cm^3$).

  Method 1 is illustrated in Fig. 3: the reactor was connected to the PTR-MS and aliquots of 0.05 μl of acetone, in a flow of zero air, were injected simultaneously at the entrance and at the exit of the reactor. Acetone was used as a tracer for these experiments because it is easy to detect with the PTR-MS at m/z 59. The simplest way to determine the residence time is to

measure the time lag between the detection of the first and the second acetone peak signals by the PTR-MS ("maxima" in





Fig. 3). However, it must be observed that the flow dynamics inside the reactor are complex and, as a consequence, there is no single value of residence time, but a distribution (Cazorla and Brune, 2010; Huang, 2016). The acetone signal from each injection can also be used to determine the mean residence time in the reactor ("means" in Fig. 3): the results differ from the residence times estimated using the "maxima" calculations by about 30%.

Method 2 uses the TORS technique to measure the reactivity of $O_3$ with NO: since the rate coefficient of $NO + O_3$ is well known ($1.73 \times 10^{-14}$ cm$^3$ molecule$^{-1}$ s$^{-1}$ at 298 K, with an uncertainty of 17% (Atkinson et al., 2004)), the only unknown variable in Eq. 3 is the reaction time $t$, in an $NO/O_3$ only reaction system under pseudo first-order conditions. The average of several experiments conducted at different flow rates is shown in Fig. 4a. Method 3 assumes perfect instant mixing and plug flow in the reactor: the calculation for a range of flow rates is shown in Fig. 4b.

All three methods offer internally consistent results, within the respective uncertainties (Fig. 4b). Method 3 agrees well with the acetone injection "means" calculation, but overestimates the residence times determined with method 2 by ∼13%. Method 2 is the preferred one, because it relies on well known kinetic parameters and it implicitly takes into account the real distribution of flow paths through the reactor. Method 2 also provides a simple test of the TORS functionality using NO instead of a BVOC. In the experiments described in Sect. 5, we used the residence time determined with method 2 with a reactor flow 275 of 2470 sccm (Sect. 4.3).

### 4.2 Ozone wall loss

One of the most important parameters and key uncertainties of the TORS technique is the ozone wall loss ($R_{\text{wall}}$), as discussed in Sect. 2.2. Other species can be lost on the reactor surface, but these are likely to have little or no impact on the determination of ozone reactivity either because they do not react with $O_3$ (e.g., multifunctional products of VOC oxidation) or because they 280 are present in too small concentrations.

Measured $R_{O3}$ is obtained by subtracting the loss of $O_3$ on the reactor wall ($R_{\text{wall}}$) and, if present, to NO ($R_{\text{NO}}$) from the total loss rate of $O_3$ (Eq. 6). Small changes in the wall loss could lead to significant variation in the final determination of ozone reactivity. It is therefore important to minimize this parameter in order to reduce the uncertainty of the measurement. In the design stage, several materials and sizes were tested under dry and humid conditions: 1) a glass cylinder (5 x 70 cm), 2) a 285 quartz cylinder (9 x 100 cm), 3) a PFA coil (1.9 x 1524 cm), 4) a PTFE cylinder (9 x 100 cm). Based on these experiments, the large diameter (OD=9 cm, ID = 8.73 cm) PTFE cylinder showed the lowest ozone wall loss and minimal dependence on humidity.

The ozone wall loss was regularly determined during measurements by switching to a flow of zero air instead of the regular sample with a 3-way teflon valve (Fig. 2): in this operating mode there are no reactants (NO or VOCs) in the reactor and the 290 measured ozone reactivity is equal to $R_{\text{wall}}$. During each experiment and measurement period, multiple determinations of the ozone wall loss were made. Figure 5 shows the average $R_{\text{wall}}$ determined during a series of laboratory experiments (Sect. 5.1 and Sect. 5.2), as a function of measured humidity, temperature and time. There was a small but noticeable dependence of $R_{\text{wall}}$ on humidity. Ambient humidity is usually higher than the range shown in Fig. 5, but because of the dilution of the sample flow by the dry ozone flow before entering the reactor (Fig. 2), the relative humidity in the reactor is always less than 50%. There





was no clear dependence of $R_{\text{wall}}$ on temperature, at least in the limited range experienced in the laboratory. The ozone wall loss can be expected to vary with time, as the surface of the reactor is passivated and exposed to ambient air, but there was no obvious temporal trend over the nine month period of these experiments (Fig. 5).

Although there was no observable pattern with respect to the measured parameters, it is apparent that there was significant variability in the ozone wall loss and, therefore, it is necessary to measure $R_{\text{wall}}$ often during an experiment and/or ambient

measurements. The interquartile range of $R_{\text{wall}}$ was $0.5 - 1.3 \times 10^{-4}\,\text{s}^{-1}$ (mean $= 1.1 \times 10^{-4}\,\text{s}^{-1}$), which corresponds to the reactivity of 22-58 ppb of $\alpha$-pinene (mean = 49 ppb). The range of the measured ozone wall losses suggests that the limit of detection of TORS is of the order of a few tens of ppb of $\alpha$-pinene or the equivalent concentration, in terms of reactivity, of other BVOCs (Table 1). The limit of detection was quantified with laboratory experiments using known concentrations of $\alpha$-pinene, as described in Sect. 5.1.

**4.3    TORS operation**

The flows in the TORS instrument are constrained by several competing factors (Fig. 2): first, the total flow (sample + ozone) must be larger than the inlet flows of both $O_3$ monitors, which are fixed at $\sim$1.4 slpm each. Second, the residence time in the reactor must be long enough to allow the ozonolysis reactions to take place to a measurable extent, but short enough that the consumption of the reactants does not become significant, so that pseudo first-order conditions are maintained (Sect. 2.2).

During the design phase of the TORS instrument several combinations of flow settings were experimented with, eventually settling on a sample flow of $\sim$2.3 slpm and an ozone flow – composed of a flow of zero air through the ozone lamp, plus a dilution flow to control the concentration of ozone and a flow of cyclohexane as OH scrubber – of $\sim$1.5 slpm (Fig. 2). These settings result in a reactor flow rate of 2470 sccm, corresponding to a residence time of 128 seconds (Sect. 4.1). Since the sample flow is mixed with the ozone flow (Fig. 2), a correction factor needs to be applied to account for the dilution of the

sample by the ozone flow. In the experiments and the measurements described in Sect. 5, the mixing ratio of ozone at the entrance of the reactor was $\sim$120 ppb. The instrument settings, such as the residence time and the ozone concentration, can in principle be varied to adjust the sensitivity of TORS for environments with different values of $R_{\text{O3}}$.

Once the flows are set, the basic operation cycle of the TORS instrument consists of 3 main steps:

1. "bypass mode" to check the agreement between the two $O_3$ monitors: the sample flow is substituted with an equal flow

320        of zero air and the reactor is bypassed for a period of approximately 15 minutes. In this mode, both monitors measure the $O_3$ concentration in zero air (i.e., with no reactants) before the entrance of the reactor, so that a correction factor can be derived if they differ. In this work, the difference between the $O_3$ monitors was checked at the start and at the end of each experiment/measurement period, every day, and was typically $\sim$1 ppb.

2. "wall loss mode" to determine $R_{\text{wall}}$ (Eq. 6): the sample flow is substituted with an equal flow of zero air inside the

325        reactor for a period of approximately half an hour. The effect of the humidity change in the reactor within such a short period of time was found to be negligible. In this work, the wall loss was determined every 2-3 hours.

3. "sampling mode", the main operation mode of the instrument with the sample flow containing BVOCs.





This procedure was followed during all the experiments and measurements described in Sect. 5.

# 5 Evaluation of TORS

The TORS instrument was tested in a series of experiments to evaluate its functionality and potential. The experiments were designed and conducted in order of increasing complexity from individual species in laboratory conditions to the complex BVOCs mixture of a horticultural glasshouse. The TORS instrument was at first tested in the laboratory using pure $\alpha$-pinene (Sect. 5.1), followed by emissions from small plants (Sect. 5.2). After these tests showed that the instrument was behaving as expected under controlled conditions, it was deployed in a glasshouse containing various aromatic plants to demonstrate that

TORS can measure total ozone reactivity under quasi-ambient conditions (Sect. 5.3).

It must be noted that the rate coefficients of $O_3$ with BVOCs span several orders of magnitude (Table 1). Therefore, for individual species in isolation, a measured ozone reactivity of (for example) $2.36 \times 10^{-5}$ s$^{-1}$ corresponds equally to 1.9 ppm of camphene, 10 ppb of $\alpha$-pinene, 74 ppb of isoprene, 80 ppt of $\beta$-caryophyllene. This affects the interpretation of $R_{O3}$ measurements by TORS, particularly if the composition of the sample is not known, as would be the case when taking ambient

measurements.

## 5.1 Laboratory experiments

Several laboratory experiments were carried out using known concentrations of a selected biogenic compound. A thermostated diffusion tube and pure $\alpha$-pinene (98%, from Sigma-Aldrich) in zero air were used to provide a constant source of BVOC. The diffusion rate was controlled by varying the temperature of the diffusion tube and determined by regularly weighing it with a

precision balance over a period of several weeks. The concentration of $\alpha$-pinene was then calculated from the diffusion rate and confirmed with direct measurements by PTR-MS (Sect. 3.2).

The values of $R_{O3}$ measured during several experiments were compared with the values calculated using the known concentrations of $\alpha$-pinene in the sample (Eq. 6). The data show reasonable agreement – within the combined uncertainties of the instrument and of the $\alpha$-pinene + $O_3$ rate coefficient (41%, Atkinson et al. (2004)) – between the calculated and measured

reactivity for $\alpha$-pinene mixing ratios larger than 40 ppb (Fig. 6a). At mixing ratios below 10 ppb of $\alpha$-pinene, the measured reactivities cannot be statistically distinguished from each other and from zero. Measured $R_{O3}$ corresponding to concentrations of $\alpha$-pinene >14 ppb are linearly correlated (r$^2$ = 0.993) with a slope of $1.33 \times 10^{-6} \pm 1.12 \times 10^{-7}$ s$^{-1}$/ppb, corresponding to the sensitivity of the instrument, and an intercept of $7 \times 10^{-5} \pm 9 \times 10^{-6}$ s$^{-1}$ (Fig. 6a).

Based on these experiments, the TORS detection limit, for a residence time of 128 seconds, can be estimated between

$4.5 \times 10^{-5}$ and $9.0 \times 10^{-5}$ s$^{-1}$, corresponding to ozone reactivities equivalent to 20-40 ppb of $\alpha$-pinene (Table 1). These values are consistent with the estimates based on the range of measured $R_{wall}$, as discussed in Sect. 4.2; the actual detection limit for a given set of measurements depends on the magnitude of the ozone wall loss, which can vary significantly (Fig. 5). These values are comparable to the detection limit of the instrument described by (Matsumoto, 2014) of $1.4 \times 10^{-4}$ s$^{-1}$, for a residence time of 57 seconds.





Some experiments were conducted without adding cyclohexane to the ozone flow (Fig. 2) to verify the effect of the OH scrubber, as discussed in Sect. 2.3. Ozonolysis of BVOCs is known to generate OH radicals with different yields (Rickard et al., 1999; Johnson and Marston, 2008) which may lead to consumption of BVOCs by OH in the reactor, thus causing underestimation of $R_{O3}$. However, the experiments where the OH scrubber was used did not show substantially different results from those where it was not used (Fig. 6b). Measured $R_{O3}$ with $\sim 30$ ppb of $\alpha$-pinene was $6.75 \times 10^{-5} \pm 2.41 \times 10^{-5}$

$s^{-1}$ with cyclohexane and $6.33 \times 10^{-5} \pm 2.12 \times 10^{-5}$ $s^{-1}$ without cyclohexane: the corresponding p-value was 0.39, indicating that the difference between the two measurements is not statistically significant. The difference between the ozone reactivities determined with and without OH scrubber was $\sim6\%$, in agreement with the modelling results, which showed that the presence of the OH scrubber affects the measurements of total ozone reactivity by 5% or less (Sect. 2.3). While this is less than the precision of the ozone monitors (Sect. 3.1) we note that, in principle, the comparison of total ozone reactivity measurements

with and without an OH scrubber can yield additional information on the speciation of the VOC mixture in the sample.

## 5.2   Plant experiments

Laboratory experiments were carried out using small aromatic plants in a controlled environment to test the TORS instrument under more realistic conditions. Three plants of lemonthyme (*Thymus citriodorus*) were enclosed in a Teflon bag – with an approximate volume of 0.1 m$^3$ – filled with a continuous flow of zero air. A halogen lamp was located over the bag and a

temperature/humidity probe (Vaisala HMP110) was inserted into the bag, together with a small fan to ensure homogeneous conditions. The natural humidity of the plants caused the humidity in the bag to rise over the course of the experiment, but relative humidity remained below 50% inside the TORS reactor due to dilution with the dry ozone flow (Fig. 2) and therefore did not affect the loss of O$_3$ on the reactor wall, as discussed in Sect. 4.2. The PTR-MS was connected to the bag to identify and quantify the BVOCs that constitute the plant emissions (Sect. 3.2). The TORS instrument and the PTR-MS sampled

continuously from the Teflon bag during the experiment, which had a duration of about 48 hours.

Figure 7 shows the ozone reactivity measurements of the lemonthyme experiment, together with the reactivity calculated using the BVOCs measurements by PTR-MS. The interquartile range of measured $R_{O3}$ was $3 - 11 \times 10^{-5}$ $s^{-1}$ for the first experiment and $7 - 10 \times 10^{-5}$ $s^{-1}$ for the second experiment, with mean values of $7 \times 10^{-5}$ and $9 \times 10^{-5}$ $s^{-1}$, respectively.

Measured ozone reactivity increased by about a factor of 2 when the lamp was turned on, due to increased emissions of all

BVOCs and, in particular, of the more reactive ones (i.e., monoterpenes and sesquiterpenes, Table 1). This is because when the lamp was switched on the temperature inside the bag increased (by $\sim$10 $^\circ$C, Fig. 7), as well as the light. Isoprene emissions are controlled by both light and temperature, but monoterpenes and sesquiterpenes emissions are mostly controlled by temperature and have an exponential response to temperature (Duhl et al., 2008; Guenther et al., 2012; Hellén et al., 2018). Therefore, the emissions of these more reactive compounds tend to increase faster than those of isoprene when temperature rises quickly.

To calculate $R_{O3}$ from the PTR-MS measurements (Sect. 3.2) using Eq. 6, a number of assumptions have to be made. The only BVOC that the Proton Transfer Reaction technique can uniquely identify is isoprene. All monoterpenes and sesquiterpenes have the same molecular weight (136.24 and 204.36 g/mol, respectively) and therefore are very difficult to distinguish from each other using a soft ionization technique (de Gouw and Warneke, 2007). The PTR-MS instrument effectively reports the





sum of monoterpenes and the sum of sesquiterpenes. To account for this problem, estimated low and high $R_{O3}$ limits were

calculated. Lemonthyme is an evergreen broadleaf plant, whose main emissions (besides isoprene) are $\alpha$-pinene, $\beta$-pinene, $\beta$-ocimene (monoterpenes) and $\beta$-caryophyllene, $\alpha$-farnesene (sesquiterpenes) (Fares et al., 2011; Guenther et al., 2012). The lower limit $R_{O3}$ estimate was calculated assuming that the measured monoterpene signal was solely due to $\beta$-pinene and that the measured sesquiterpene signal was solely due to $\alpha$-farnesene. The higher limit $R_{O3}$ estimate was calculated assuming that the measured monoterpene signal was solely due to $\beta$-ocimene and that the measured sesquiterpene signal was solely due to

$\beta$-caryophyllene. This provides a range of $R_{O3}$ which likely includes that of the particular BVOCs mixture emitted by the lemonthyme plants (Table 1). The calculated low and high $R_{O3}$ limit estimates are compared to the TORS measurements in Fig. 7. The TORS measured reactivities were within the range of these estimates and followed the same pattern, with higher values when the light was on and the temperature higher.

### 5.3 Glasshouse experiments

In order to evaluate the TORS technique under quasi-ambient conditions, the instrument was deployed in a horticultural glasshouse containing a range of aromatic plants. The glasshouse is a similar environment to ambient and was subject to a continuous inflow of ambient air, but, being a semi-enclosed system, the concentrations of BVOCs emitted from the plants are higher and the concentrations of NO lower than the external environment, resulting in a stronger $R_{O3}$ signal. The glasshouse is located at the Winterbourne House and Garden (https://www.winterbourne.org.uk/), adjacent to the University of Birming-

ham campus, and has an approximate volume of $200\,\mathrm{m}^3$. The following plants were inside the glasshouse during the sampling period: Fringed "French" Lavender (Lavandula dentata var. candicans), Lemon Verbena (Aloysia triphylla), Scented Leaf Geranium (Pelargonium "Prince of Orange", Pelargonium "Radula") and several varieties of Citrus (Citrus x limon, Citrus "Tahiti", Citrus reticulata "Clementine").

The TORS instrument was setup in a similar way as in the plant experiments (Sect. 5.2), with regular determination of the

ozone wall loss using a flow of zero air instead of the ambient flow. The measurements were taken over a period of two weeks in early June 2018; during this period the weather was dry (mean RH = 57%) with temperatures reaching a maximum of 39 °C inside the glasshouse (mean = 15 °C). Cyclohexane was used as OH scrubber only during the second week of measurements. The PTR-MS was not available at the glasshouse, but two air samples were taken on two different days using adsorption tubes and qualitatively analyzed by GC-MS (Sect. 3.2). The GC data were used to determine the most important monoterpenes and

sesquiterpenes in the air inside the glasshouse, based on their relative abundance.

The ozone reactivity measurements made in the glasshouse are shown in Fig. 8. For the period without cyclohexane the interquartile range was $2.11-5.01\times10^{-4}\,\mathrm{s}^{-1}$ with a mean of $3.63\times10^{-4}\,\mathrm{s}^{-1}$. For the period with cyclohexane the interquartile range was $2.01-4.09\times10^{-4}\,\mathrm{s}^{-1}$ with a mean of $3.07\times10^{-4}\,\mathrm{s}^{-1}$. Taking into account the natural variability of plant emissions, these numbers suggest that the use of an OH scrubber does not change significantly the TORS measurements, in keeping with

the laboratory experiments (Sect. 5.1) and the model results of the chemistry inside the reactor (Sect. 2.3).

In the absence of BVOC measurements, an estimate of the ozone reactivity was calculated using the qualitative information obtained from the GC-MS analysis of the adsorption tubes and emission factors from Guenther et al. (2012): broadleaf





evergreen plants emit isoprene, monoterpenes and sesquiterpenes in a proportion of approximately 1:0.1:0.02. However, Fares
et al. (2011) found that Citrus plants, several types of which were present on the glasshouse, emit more monoterpenes than

isoprene. Since most monoterpenes are more reactive with ozone than isoprene (Table 1), the estimates of $R_{O3}$ discussed below
are relatively insensitive to the actual isoprene concentration.

Analysis of the adsorption tubes showed that the most important monoterpenes were limonene, $\beta$-pinene, camphene, myrcene
and the most important sesquiterpenes were longifolene and farnesene. Based on these results, a high $R_{O3}$ was estimated as-
suming 100 ppb of isoprene, 25 ppb of myrcene and 5 ppb of $\alpha$-farnesene. A low $R_{O3}$ was estimated assuming 50 ppb of

isoprene, 5 ppb of camphene and 1 ppb of longifolene. The estimated low and high $R_{O3}$ are of the same magnitude as the
TORS measurements (Fig. 8).

In contrast to the laboratory experiments, which were performed using zero air (Sect. 5.1 and Sect. 5.2), the ozone reactivity
measurements in the glasshouse were affected by ambient NO (R5). Measurements of NO were not available in the glasshouse,
but ambient data from a nearby site can be used to estimate the role of NO. These observations were taken at the Automatic Ur-

ban and Rural Network monitoring site of Birmingham Acocks Green, which is located $\sim$7 km from Winterbourne House and
is considered a urban background site (https://uk-air.defra.gov.uk/networks/site-info?site_id=AGRN). The mean NO mixing
ratio reported at the Acocks Green site during the measurement period was 2 ppb (interquartile range = 1-2.6 ppb).

The average total ozone reactivity showed a clear diel pattern (Fig. 9), with maximum values of about $6 \times 10^{-4}\,\mathrm{s}^{-1}$ around
06:00 (approximately one hour after dawn) and was anticorrelated with ambient temperature (Fig. 8). BVOCs emissions are

driven by both light and temperature and are therefore higher during the day (Fares et al., 2011; Hellén et al., 2018). Likewise,
NO concentrations are higher during day, due to traffic emissions. Therefore, it may be expected that measured $R_{O3}$ (from
both BVOCs and NO) is higher during the daylight hours, which was in fact observed during the laboratory experiments with
the lemonthyme plants (Fig. 7). However, under ambient conditions, BVOCs react during the day with OH radicals at a faster
rate than they react with $O_3$ (Atkinson et al., 2006; Johnson and Marston, 2008). As a result, ozone reactivity tends peak in

the early morning (Fig. 9), when the NO and BVOCs emissions start increasing, but the concentration of OH is still too low to
compete with $O_3$ for BVOC removal (Hellén et al., 2018).

An assessment of the contribution of NO to the total ozone reactivity has to take into account a number of factors. First, it
must be considered that, since the NO + $O_3$ reaction is very fast (Table 1), a fraction of ambient NO is removed by ambient
ozone before it enters the TORS reactor. The inlet residence time is $\sim$4 seconds and the lifetime of NO with 20-30 ppb of $O_3$

(typical ambient levels) is 3-5 seconds, which means that the concentration of NO drops to $\sim$36% of the ambient concentration
while inside the inlet line (Fig. 2). Additionally, it may be assumed that the concentration of NO inside the glasshouse was
lower than outside, especially considering that the glasshouse is located 50 m or more from nearby roads. Therefore, under
the assumption that the concentration of NO in the reactor is $\sim$20% of the ambient concentration, $R_{NO}$ can be subtracted
from the total $O_3$ reactivity and $R_{O3}$ can be derived from Eq. 6. Based on these approximations, the ozone reactivity due to

BVOCs in the glasshouse can be estimated at $2-5 \times 10^{-4}\,\mathrm{s}^{-1}$, on average. Figure 9 shows that NO reactivity dominates during
the day, but is not significant during the night when BVOCs control ozone loss. Caution should be exercised in interpreting



these measurements and results, as the assumption about NO levels in the glasshouse could not be verified at the time of the measurements.

## 6 Summary and Future Work

An instrument to measure total ozone reactivity, named Total Ozone Reactivity System (TORS), was developed, characterized and tested under controlled conditions in the laboratory; both individual compounds and small plants were used. The instrument was deployed inside a horticultural glasshouse containing a range of aromatic plants to evaluate its functionality under quasi-ambient conditions.

The TORS instrument was able to measure $O_3$ reactivities with BVOCs ($R_{O3}$) of $4.5-9.0\times10^{-5}\,\mathrm{s}^{-1}$ or more, corresponding
to 20-40 ppb of $\alpha$-pinene, 150-300 ppb of isoprene or 160-320 ppt of $\beta$-caryophyllene. These mixing ratios are larger than typical ambient levels, but can be found in enclosure studies (Duhl et al., 2008; Bouvier-Brown et al., 2009) and can easily be reproduced in laboratory and environmental chamber experiments. An OH scrubber (cyclohexane) was used to remove the OH radicals formed by the ozonolysis of BVOCs; however, simulations of the chemistry inside the TORS reactor using a Master Chemical Mechanism (MCM v3.3.1) box-model found that the formation of OH from BVOC + $O_3$ reactions affected
the measurements of $R_{O3}$ by <5%, under the conditions used during the experiments.

Further work will improve the stability of the signal, reduce the signal noise and the detection limit. This may require using ozone monitors with higher precision and/or a more stable $O_3$ generator, as well as a detailed exploration of the various parameters affecting TORS: gas flows, residence time, relative humidity, OH scrubber levels, ozone concentrations. In the future, TORS will be able to make ambient measurements, especially in environments with high BVOC loading, to improve
understanding of the role of natural emissions on the ozone budget and the oxidative capacity of the atmosphere.

*Competing interests.* The authors declare no competing interests.

*Acknowledgements.* Thanks to D. Blenkhorn, V. Matthaios, K. Vohra, D. Komalasari for their help with some of the experiments and to the University of Birmingham Biosciences Workshop for their continuous assistance. Special thanks to the staff of the Winterbourne House and Garden for their support and to Prof. Jun Matsumoto for his advice and kind hospitality. Funding provided by NERC (NE/P003524/1) and
the UoB-Waseda Research Collaboration Fund.



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



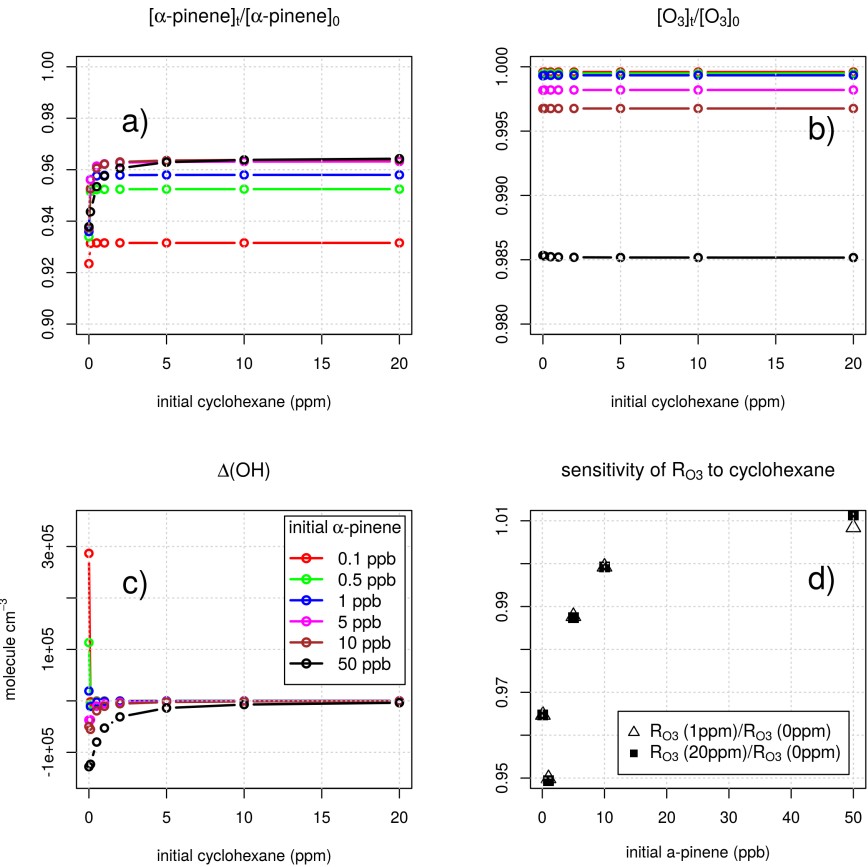

**Figure 1.** Modelled removal of $\alpha$-pinene and $O_3$ (relative to their initial concentrations) as a function of cyclohexane level (a-b). Modelled difference between the initial and final concentrations of OH as a function of cyclohexane level (c). Sensitivity of modelled $R_{O3}$ to cyclohexane concentrations as a function of the initial $\alpha$-pinene concentration (d).



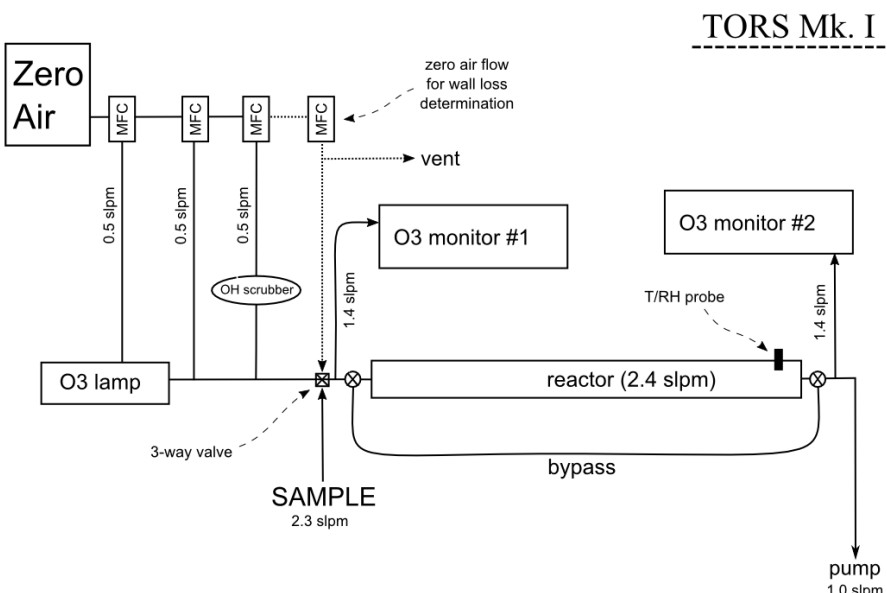

**Figure 2.** Diagram of the TORS instrument with typical flow settings (Sect. 4.3).



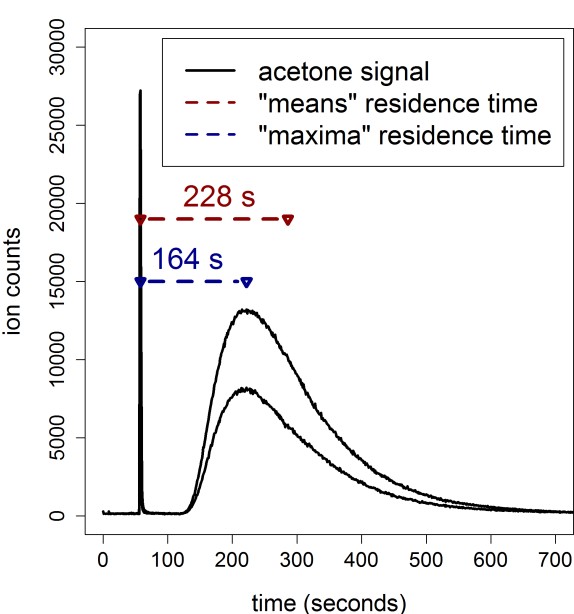

**Figure 3.** Two acetone double injection experiments used to estimate the residence time in the PTFE reactor for a flow rate of 1600 sccm. The "maxima" calculation uses the differences in time of the peak acetone signals; the "means" method uses the differences between the mean elapsed time of the two acetone signals.

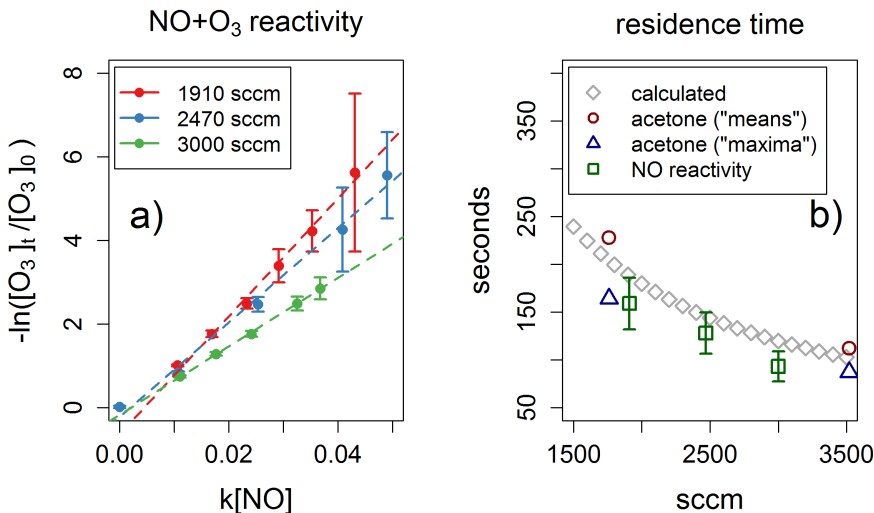

**Figure 4.** NO reactivity experiments (a). Residence times as a function of reactor flow, determined by three different methods (b). The results of the acetone injection method are taken from Fig. 3.



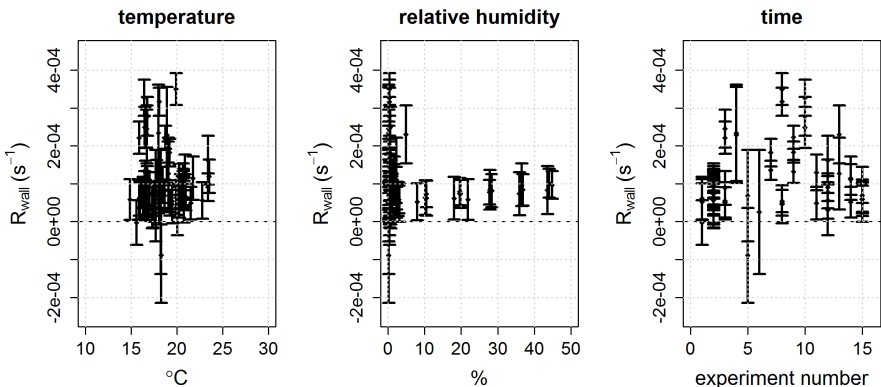

**Figure 5.** Mean and 2-$\sigma$ standard deviation of the ozone wall loss ($R_{\text{wall}}$) in the PTFE reactor as a function of humidity, temperature and time. The data are taken from 15 experiments, with multiple determinations of $R_{\text{wall}}$ per experiment, over a nine month period.

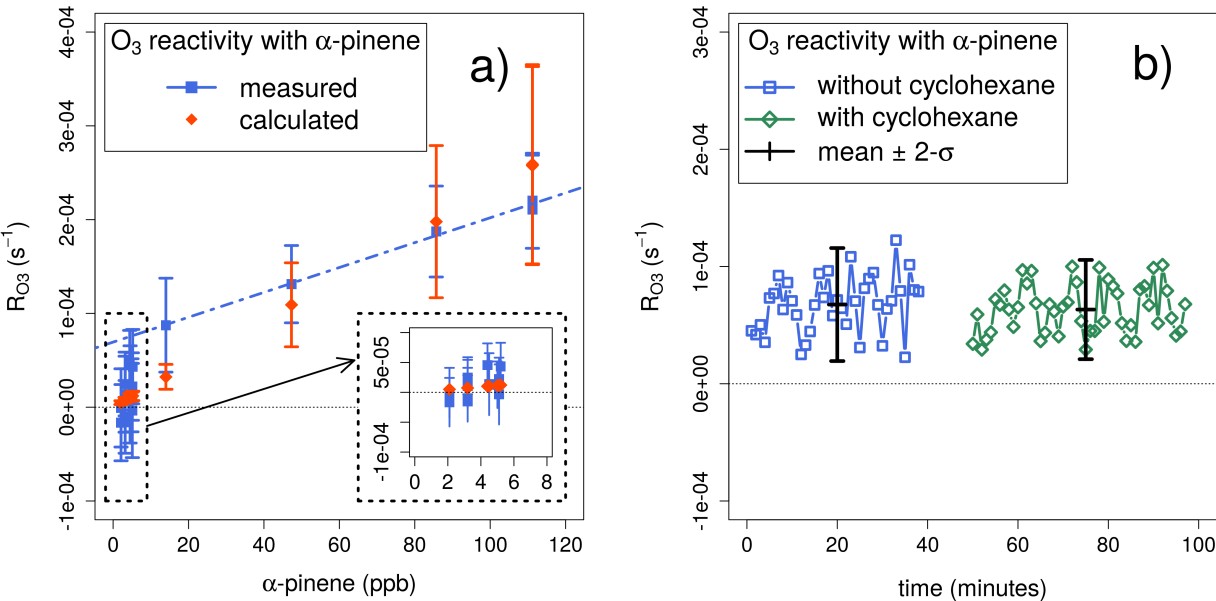

**Figure 6.** Measured and calculated mean ozone reactivities with $\alpha$-pinene (a). Ozone reactivity with $\alpha$-pinene measured with and without cyclohexane as OH scrubber (b). The blue dashed line (left panel) indicates the linear regression of the measured $R_{O3}$ values for $\alpha$-pinene mixing ratios >20 ppb.

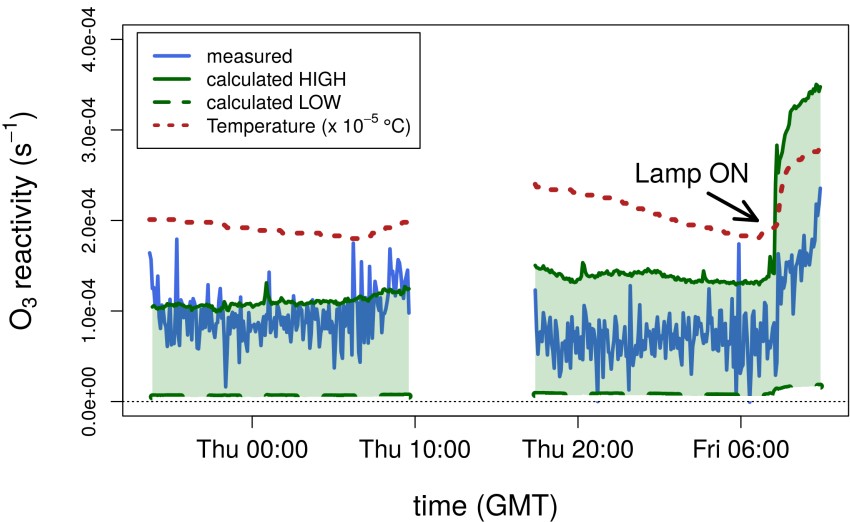

**Figure 7.** Measured ozone reactivity from 3 plants of lemonthyme (*Thymus citriodorus*) in a Teflon bag, compared to the ozone reactivity calculated from measurements of BVOCs by PTR-MS. The temperature (in °C) measured inside the bag is also shown. All data are averaged to 5 minutes.

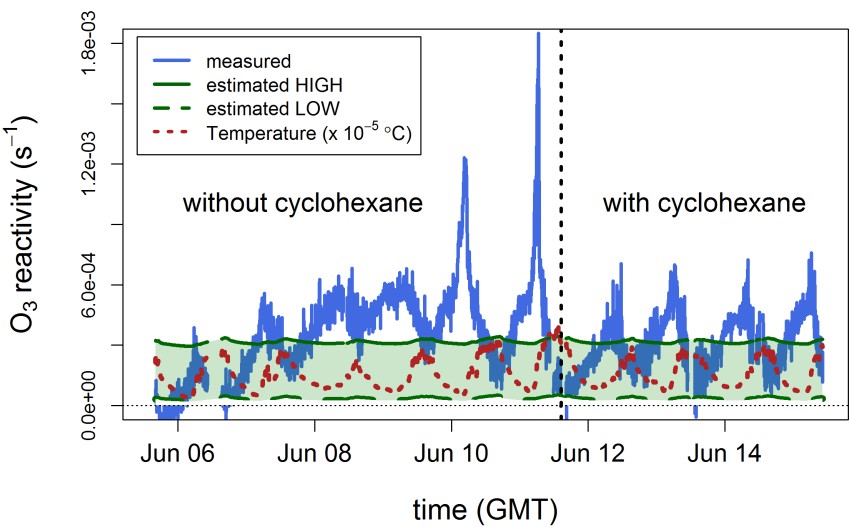

**Figure 8.** Measured ozone reactivity in the Winterbourne House and Garden glasshouse compared to ozone reactivity calculated from estimated BVOC concentrations. The temperature (in $^\circ$C) measured inside the glasshouse is also shown. All data are averaged to 5 minutes.



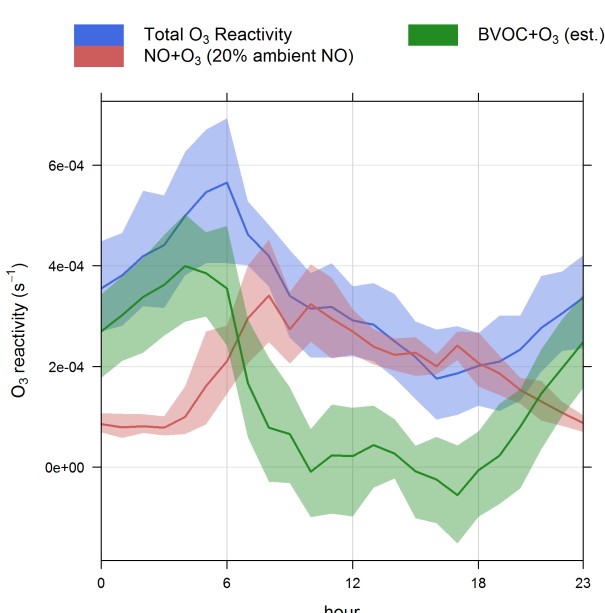

**Figure 9.** Diel averages of measured total ozone reactivity and estimated ozone reactivities with NO ($R_{NO}$) and BVOCs ($R_{O3}$).



**Table 1.** Rate coefficients of the reactions with ozone ($k_{O3}$, in $cm^3$ molecule$^{-1}$ s$^{-1}$), absolute ozone reactivities ($R_{O3}$, in s$^{-1}$) and relative ozone reactivities (with respect to $\alpha$-pinene) of selected BVOCs and inorganic compounds. The rate coefficients are from Atkinson et al. (2004, 2006) and calculated for standard conditions (298 K, 1 atm); the ozone reactivities are calculated for mixing ratios of 1 ppb.

| Class | Species | $k_{O3}$ | $R_{O3}$ | relative $R_{O3}$ |
|---|---|---|---|---|
| hemiterpenes | isoprene | $1.28 \times 10^{-17}$ | $3.15 \times 10^{-7}$ | 0.13 |
| monoterpenes | $\alpha$-terpinene | $1.90 \times 10^{-14}$ | $4.68 \times 10^{-4}$ | 197.97 |
| | $\beta$-ocimene | $5.70 \times 10^{-16}$ | $1.25 \times 10^{-5}$ | 5.31 |
| | myrcene | $4.70 \times 10^{-16}$ | $1.16 \times 10^{-5}$ | 4.90 |
| | limonene | $2.20 \times 10^{-16}$ | $5.41 \times 10^{-6}$ | 2.29 |
| | $\alpha$-pinene | $9.60 \times 10^{-17}$ | $2.36 \times 10^{-6}$ | 1 |
| | $\beta$-pinene | $1.89 \times 10^{-17}$ | $4.67 \times 10^{-7}$ | 0.20 |
| | sabinene | $8.30 \times 10^{-17}$ | $2.04 \times 10^{-6}$ | 0.86 |
| | 3-carene | $4.90 \times 10^{-17}$ | $1.21 \times 10^{-6}$ | 0.51 |
| | camphene | $5.02 \times 10^{-19}$ | $1.24 \times 10^{-8}$ | 0.01 |
| sesquiterpenes | $\beta$-caryophyllene | $1.20 \times 10^{-14}$ | $2.96 \times 10^{-4}$ | 125.03 |
| | $\alpha$-farnesene | $5.88 \times 10^{-16}$ | $1.45 \times 10^{-5}$ | 6.13 |
| | $\alpha$-copaene | $1.50 \times 10^{-16}$ | $3.69 \times 10^{-6}$ | 1.56 |
| | longifolene | $5.00 \times 10^{-19}$ | $1.23 \times 10^{-8}$ | 0.01 |
| inorganic | NO | $1.89 \times 10^{-14}$ | $4.65 \times 10^{-4}$ | 196.56 |
| | $HO_2$ | $2.01 \times 10^{-15}$ | $4.96 \times 10^{-5}$ | 20.99 |
| | $NO_2$ | $3.52 \times 10^{-17}$ | $8.67 \times 10^{-7}$ | 0.37 |