# Peer review of "An instrument for in situ measurement of total ozone reactivity"

_Atmospheric Measurement Techniques, 2019_

## Referee Comment (RC1) · Anonymous Referee #1 · 30 Aug 2019

The authors present a description of an instrument to measure total ozone reactivity, details of experiments to characterise the instrument and initial results from measurements of the total ozone reactivity made using individual compounds, emissions from a single plant species, and in a glasshouse from several plant species. The manuscript demonstrates the potential for measurements of total ozone reactivity, but is lacking detail in some areas which should be addressed prior to publication. Specific comments are provided below.

Abstract: The accuracy, time resolution and limit of detection (and corresponding integration time) should be clearly stated in the abstract.

Page 1, line 16: Consider changing 'atmosphere' to 'troposphere'.

[Figure]

Page 1, line 17 (and elsewhere): Provide the relevant wavelengths for the reaction.

Page 2, line 29: Change '... state and primarily reacts ...' to '... state which primarily reacts ...'.

Page 3, line 52: The statement 'all BVOCs are very reactive with both OH and O3 ...' is not really true. At the top of the page, methanol, CO and acetone are listed as significant BVOC emissions, none of which are very reactive with O3.

Page 4, line 78: ki is the bimolecular rate coefficient, not the pseudo-first-order rate coefficient.

Page 4, line 85: Note that measurements of HO2 reactivity have also been reported (Miyazaki et al., Rev. Sci. Instr., 84, 7, doi.10.1063/1.4812634).

Page 4, line 88: Are there other considerations for long-lived species? Is it necessary to assume that O3 is in steady state?

Page 4, line 97: Consider changing 'when photolysis is zero' to 'when photolysis rates are zero'.

Page 4, line 99: The comparisons between RO3 resulting from NO2, alpha-pinene and limonene are a little confusing. If NO2 has the lower rate coefficient it should require a greater concentration to reach the same O3 reactivity as alpha-pinene or limonene. For the rate coefficients given in Table 1, and assuming T = 298 K, p = 1 atm, 1 ppb of NO3 has RO3 = 8.7e-7 s-1, 2.7 ppb of alpha-pinene has RO3 = 6.4e-6 s-1 and 6.2 ppb of limonene has RO3 = 3.4e-5 s-1. Should this read that 2.7 ppb of NO2 has the same RO3 as 1 ppb of alpha-pinene and 6.2 ppb of NO2 has the same RO3 as 1 ppb of limonene? Please clarify.

Page 5, line 127: The previous study describing measurements of RO3 defines it as the total O3 reactivity, and in their experiments/measurements assume [NO] = 0 and all observed RO3 is a result of reactions with VOCs. In this case, where NO is present and its effects on the observed RO3 has to be subtracted to give the O3-VOC reactivity,

would it be sensible to define this as a separate parameter to RO3 where the O3-NO reactivity is known? This would avoid any future confusion between studies that may define RO3 as the total observed reactivity (as in the previous work) or as the subset of RO3 owing to O3-VOC reactivity (as in the current work).

Page 6, line 149: Please quantify the statement 'not substantially different'.

Page 7, line 169: Spelling of 'independent'.

Page 8, line 210: Please provide some further details regarding the requirements for the residence time. What difference in [O3] is required for accurate measurements of ozone reactivity? How much change in [VOC] is acceptable before the measurement of ozone reactivity is affected?

Page 8, line 224: There is no hyphen in 'ad hoc' (also similar comments for in situ, 2 sigma, elsewhere).

Page 9, line 224: Please provide details of the mass transmission curve. What is the source? How does it affect the uncertainties in the measurements? What are the limits of detection for the VOC measurements?

Page 9, line 259: What is 'easy' about detecting the peak at m/z 59? Mass separation from other peaks? Peak height/ionisation cross-section for acetone compared to other compounds?

Page 10, line 261 onwards: What is the flow regime in the instrument? Is the assumption of plug flow appropriate? How was the concentration of NO determined in method 2?

Page 10, line 292: Please quantify the 'small but noticeable dependence of Rwall on humidity'.

Page 11, line 295: State the temperature range in the text.

Page 11, line 298 onwards: What is the standard deviation and median of the mea-

sured ozone wall loss? Can the limit of detection be quantified more precisely using the observed variability in the ozone wall loss?

Page 11, line 311: Please provide further details of the experiments that led to '... eventually settling on a sample flow of ~2.3 slpm'. What were the ranges of conditions investigated? How did the instrument perform under these conditions? Why was a flow of ~2.3 slpm considered optimum?

Page 11, line 323: What is the impact of the difference of ~1 ppb on the uncertainty in the ozone reactivity measurements?

Page 12, line 345: How well did the concentrations determined from the diffusion tubes agree with those determined by the PTR-MS measurements?

Page 12, line 348: Please quantify 'reasonable agreement'.

Page 12, line 358: The range of values for the limit of detection are 1/3 to 2/3 of that described previously, 'comparable' is somewhat subjective. How does the residence time affect this? A more detailed description of the instrument used in previous studies would be helpful to provide the reader with a more informed comparison.

Page 13, lines 372-375: What were the sources of the Teflon bag, halogen lamp and small fan?

Page 13, line 376: What is meant by 'the natural humidity of the plants'? Natural release of water vapour via transpiration and evaporation by the plants?

Page 13, line 383: What are the uncertainties in the stated values?

Page 14, line 408: Please provide some approximate quantification for the statement 'concentrations of BVOCs ... are higher and the concentrations of NO lower ...'.

Page 14, line 415: What was the variability in the measured wall loss?

Page 14, line 420: What were the most important species?

Page 14, lines 422 and 423: What are the uncertainties in the stated means?

Page 15, line 449: Change '. . . ozone reactivity tends peak . . .' to '. . . ozone reactivity tends to peak . . .'.

Page 15, line 458: What is the basis for the assumption of NO in the reactor being $\sim$20 % of the ambient concentration?

Page 16, line 476: It's not clear how the listed improvements will be achieved or how TORS will be able to make ambient measurements (line 479). Specific details would help to avoid this simply reading as a wishlist.

---

## Referee Comment (RC2) · Anonymous Referee #2 · 3 Sep 2019

A. General comments:

In this manuscript, authors focused on total ozone reactivity, built and characterized their instrument in the laboratory, and tested it in the glasshouse. This study is positioned as a basic research of the instrument and a demonstration for measuring BVOCs emission from plants. Ozone reactivity of BVOCs is interesting for investigating BVOCs. Thus, the reviewer believes that this work has an important implication and is significant enough to be published in this journal. However, the present manuscript leaves several points to be improved, clarified, modified, and/or reconstructed, in order for readers to understand descriptions and to recognize the significance of this study clearly. Especially, it is necessary to correct critical errors on quantitative descriptions, to indicate more information and explanations, and to clarify the story of discussion which authors

want to express during the demonstration of the instrumental test.

B. Important specific comments:

B1) Line 101 and followings: Quantitative descriptions on contribution of NO2 should be corrected. The reviewer thinks that 1 ppbv of NO2 corresponds to 3.52/9.6 = 1/2.7 = 0.37 ppbv of a-pinene and 3.52/22 = 1/6.25 = 0.16 ppbv of limonene, respectively. And then, consequently, based on the correct values, descriptions on importance of nighttime NO2 should be checked again, including descriptions in Sect. 5. Please do not mislead readers.

B2) Fig.1 and Lines 160 $\sim$ (descriptions on model estimations) : Please show more information and descriptions on model estimations. For example, how much is the initial OH concentration? Why does the delta-OH in Fig.1 distribute less than zero? At all, what is 'delta-OH' in Fig.1? (No descriptions and explanations in sentences.) Additionally, please add descriptions on the applied reaction time in the sentence and in the caption of Fig.1.

B3) Line 204: Why and how can 'ambient temperature and pressure' affect the chemistry inside the reactor? Now there are no explanations in the manuscript.

B4) Line 258 'simultaneously' : How did authors check and ensure the simultaneity? Uncertainties on the timing of the synchronized injection of acetone can cause uncertainties on the determined reaction time.

B5) Fig.4 & Sect.4.1 (especially, on NO+O3 reaction): In 'Method 2 (Fig.4a)', k[NO] up to 0.05 (maybe in the unit 's-1'? please clarify the unit;) were adopted. It means that [NO] up to about 100 or 110 ppbv, I think. I want to know authors' opinions on following questions and for authors to revise the descriptions to clarify the situations: [Q1] Concentrations of NO and O3 are similar. The reaction NO+O3 is fast. Then, both NO and O3 can significantly decrease within the reaction time. (Question:) Are the settings of the experiments, calculations, and discussion to determine the reaction

time proper to realize and ensure the 'pseudo first-order reaction' as described in Line 267 ? When I tried a rough and simple calculation on the decrease of O3 and NO, for example, till t = 150 sec, NO decreased from 100 ppbv to 5 ppbv and O3 decreased from 120 ppbv to 25 ppbv. NO can drastically decrease during the reaction time due to its fast reaction with O3. [Q2] In Fig.4a, we can estimate the reaction time. For example, in case of 2470 sccm, for k[NO] = 0.04 (s-1?), -LN(O3(t)/O3(0)) is about 4, and the slope of the regression line is roughly found as 4/0.04 $\sim$ 100 sec. However, for 2470 sccm in Fig.4b, we can read out the reaction time determined by NO reactivity as about 130 sec. Which figures are correct? And is the reaction time that authors determined and described in the manuscript exactly correct? These are critical points because the reaction time is one of the most important factors to determine ozone reactivity. Would you please confirm them and, as necessary, revise descriptions in order to clarify authors' standpoints? Additionally, as an associated question, what value of k(NO+O3) was applied in this study? Authors indicate the values as '1.89x10-14' in Table 1 and '1.73x10-14' in Line 266. The ratio 1.89/1.73 is 1.09. If authors mistake the values of k, it results in $\sim$ 10 % errors by itself.

B6) Fig.6a and Sect. 5.1.: Between calculations and experiments, the dependence on a-pinene concentrations can be seen different. Experimental data rise up steeply (a-pinene < 10 ppb) and then gradually increase (with smaller slope than calculations). Calculations show a straight line. Would you please explain briefly these results in the sentences? What happens in the reactor, do you think?

B7) Sect. 5.3, about NO contribution to ozone reactivity: It is unreasonable to understand that the present descriptions that NO concentration in the house is assumed as 20 % of ambient concentration (7 km far from the experimental site). We think that the assumption on NO concentration is conveniently and arbitrarily made. If authors have any information to validate it, it is necessary to show the evidence and to explain them clearly. If not, it is necessary to indicate clearly authors' opinion, procedures of analysis, flow of argument, and the positioning of this experiment. Especially, please

distinguish between the fact and the interpretation. For example, the reviewer's understanding of this section is as follows . . . (Fact) This experiment is aimed at a challenging demonstration of the TORS instrument. The major point is ozone reactivity measurement. It is true that significant RO3 was captured during the experiment. (Assumption & Caution) Authors want to discuss on BVOCs contribution to captured RO3 data. To do it, NO contribution should be considered. However, NO concentration in the sample is not monitored. Then the monitoring data (7km far) were adopted in order to know rough information on NO concentration. However, NO concentration can vary largely in the urban area due to the location, traffics, time of day, etc. Thus, adopted NO concentration is arbitrary and has large uncertainties. (It is unfair if descriptions on the limitations are insufficient.) (Interpretation & Authors' opinion) (For example) To discuss NO contribution, NO concentration was assumed as 0 to 100 % of ambient (7km far) concentration. When NO was set to ?? % or more, ozone reactivity by NO is larger than captured RO3. So it was suggested that NO concentration was less than ?? % of ambient level. In this study, as an upper limit (?), NO was assumed as 20 % of ambient level where NO contribution is equal to and/or less than captured RO3 (Fig.9). Then BVOCs contribution to RO3 was indicated as ?? in Fig.9. . . . Other associated comments are shown as follows: B7-2) Line 441 'urban background site' and Line 457 '50m from nearby roads': Please clarify the positioning of each site. Is the background site far from roads (i.e. not 'roadside')? Is the experimental site also considered as 'urban background'? B7-3) Fig.8: NO concentration data at the urban background site (or NO contribution to RO3) should be indicated, because they are essential for us to discuss NO contribution to RO3 in Fig.9. For example, does NO really show the daytime peak as Fig.9? Does NO indicate its variation similar to RO3? B7-4) Such authors' standpoint and assumptions should be noted at the beginning of the paragraph, as well as at the end. 'This is only a (challenging?) rough estimation & interpretation based on some assumptions', 'NO concentration is not monitored; NO contribution is assumed', for example. Also in the caption of Fig.9, such descriptions on 'assumption' are desirable.

B8) Sect. 5.3, about O3 and NO2 in the glasshouse: Authors consider that NO concentration in the house is smaller than ambient. Then, how about O3 and NO2 concentrations in the house? Is the glasshouse enclosed by walls? Or, can ambient air pass through the house? As a result, how much are O3 and NO2 concentrations in the house, do you think? Are O3 and NO2 in the house significant for RO3 measurement?

B9) Fig.9, about the diurnal variation of BVOCs contribution to RO3: Why does the BVOCs contribution indicate their peak during night (or before dawn)? Because BVOCs emission from plants usually depend on temperature and light intensity, it is expected that BVOCs contribution to RO3 has daytime peak. Would you please add explanations and your opinions on such diurnal variation of BVOCs contribution?

B10) Around Line 455, about the lifetime of NO by O3 reaction: k is about 2 x 10ˆ-14, [O3] (30 ppb) is about 7.5x10ˆ11, then k[O3] is about 1.5x10ˆ-2 s-1. Thus, the lifetime of NO is about 67 sec ( = 1/(1.5x10ˆ-2 s-1) ). This error is critical for the authors' consideration that NO can be reduced to 20 % of ambient level due to O3 reaction in the sample tube (4 sec). Associated to the comment B8, can ambient NO react with residual O3 in the glasshouse before the sampling inlet? Please consider them again and reconstruct the descriptions.

C. Other comments and Technical corrections:

C1) Line 29: 'state' —> states' ?

C2) Lines 64 - 65 and References: If authors want to refer the proceeding of a conference (Park et al.(2013), another earlier proceeding in the previous conference should be referred: Matsumoto, J.: Comprehensive analyzer for biogenic volatile organic compounds detected as total ozone reactivity, in AGU 2011 Fall Meeting, 2011.

C3) Line 71 'known BVOC mixture' —> The reviewer cannot find descriptions on such 'mixture' in the following sections. Please clarify what the mixture is. (e.g. mixture of a-pinene and cyclohexane? but the scavenger is not BVOC . . . )

[Figure]

C4) Line 90 (eq.2) and Line 105 (eq.3): Ozone concentrations, [O3], are missing in all loss terms of ozone.

C5) Line 91 'R11' —> 'R2', too?

C6) Line 131 'R5-R8' —> 'R6-R8'?

C7) Line 140 'Sect.4.2 —> Sect.5.1, too?

C8) Line 146: Please add brief descriptions on 'Sommariva et al., submitted'.

C9) Lines 178-184 and Fig.1d: In Fig.1d, RO3(with)/RO3(w/o) (please add the axis name in the figure) are $\sim$ 0.965 at a-pinene = 0.1 ppb and $\sim$0.950 at 0.5 ppb. That is, RO3(w)/RO3(wo) ratio decreases while a-pinene < 0.5 ppb, and then increase for a-pinene > 0.5 ppb. Would you please add brief explanations on this trend?

C10) Line 193 'ambient measurements' —> Strictly, 'ambient, not always, but nighttime & high NOx & low NO' ? Please clarify the conditions.

C11) Line 264 'about 30 %': It is true that '164 s' is 70 % of '228 s' and thus 30 % smaller than '228 s'. However, '228 s' is 139 % of '164 s' and thus about 40 % larger than '164 s'. The descriptions are not clear, including the word 'difference'.

C12) Line 267 'Eq.3' is not correct. the reviewer recommends that this manuscript be acceptable after minor revisions.

C13) Around Line 294: Please clarify, anywhere in the sentences earlier, what the 'central? 0.5 slpm flow of zero air (without O3 lamp and scrubber)' in Fig.2 means. Then the mean of 'dilution' (Line 293) can be clear. (The zero flow may be used as 'dilution flow to control the concentration of ozone' as described in Line 312. However, before Line 294, we have not already recognized that point.)

C14) Fig.8 'estimated HIGH & LOW': Why do these data indicate diurnal variation? How to determine these data?

C15) Scientific names of plants: Italics?

C16) Line 470: It is desirable when conditions like averaging time and reaction time are also indicated.

C17) Table 1: If possible, please indicate the lifetimes of VOCs for O3 = 120 ppbv. Then we can compare the lifetimes to reaction time and discuss the pseudo first-order reaction.

C18) Figures: Please add the names (titles) of axis.

End of Comments.

---

## Author Comment (AC1) · 7 Jan 2020

**Response to Anonymous Referee #1**

The authors present a description of an instrument to measure total ozone reactivity, details of experiments to characterise the instrument and initial results from measurements of the total ozone reactivity made using individual compounds, emissions from a single plant species, and in a glasshouse from several plant species. The manuscript demonstrates the potential for measurements of total ozone reactivity, but is lacking detail in some areas which should be addressed prior to publication. Specific comments are provided below.

We thank the referee for the detailed comments and suggestions. Below are our responses and related modifications to the manuscript. The line numbers refer to the version of the manuscript published on AMTD.

Abstract: The accuracy, time resolution and limit of detection (and corresponding integration time) should be clearly stated in the abstract.

The estimated limit of detection is already stated in the abstract (line 5). The integration time (5 minutes) has now been added. By propagating the uncertainties in the determination of the residence time (which depend on the uncertainty of the NO+O3 reaction kinetics and of the NO concentration), and in the ozone measurements, and by taking into account the typical variability of the ozone wall loss, we estimate the total uncertainty of the instrument to be about 32%, based on equations 5 and 6. Note however that both the uncertainty and the detection limit of the instrument are somewhat variable because they are affected by the loss of ozone on the reactor wall – this element has been incorporated (conservatively) in the uncertainty estimate (see discussion in Section 4.2). The following modifications were made to the abstract, to the conclusions and to Section 5.1:

Lines 5-6: changed to "and proved able to measure reactivities corresponding to > 4.5e-5 s-1 (at 5 minutes averaging time), with an estimated total uncertainty of ∼32%. Such reactivities correspond to >20 ppb of a-pinene or >150 ppb of isoprene in isolation"

Line 359: added "From Eq. 5 and Eq. 6, the total uncertainty of TORS can be estimated by propagating the uncertainties in the determination of the residence time (related to the uncertainty in the concentration of the NO cylinder and in the rate coefficient of the NO+O3 reaction, Sect. 4.1), the uncertainty of the ozone monitors (Sect. 3.1) and the median variability of R(wall) during individual experiments (Sect. 4.2) at ∼32%."

Lines 469-470: changed to "The TORS instrument was able to measure O3 reactivities with BVOCs (R(O3)) of 4.5-9.0e-5 s-1 or more – with a residence time of 140 seconds, an averaging time of 5 minutes, and an estimated total uncertainty of ∼32%. These values correspond to 20-40 ppb of a-pinene, 150-300 ppb of isoprene or 160-320 ppt of β-caryophyllene."

Page 1, line 16: Consider changing 'atmosphere' to 'troposphere'.

Done.

Page 1, line 17 (and elsewhere): Provide the relevant wavelengths for the reaction.

Done.

Page 2, line 29: Change '. . . state and primarily reacts . . .' to '. . . state which primarily reacts . . .'.

Done.

Page 3, line 52: The statement 'all BVOCs are very reactive with both OH and O3 . . .' is not really true. At the top of the page, methanol, CO and acetone are listed as significant BVOC emissions, none of which are very reactive with O3.

Changed to "Most BVOCs are reactive with OH and many of them, such as isoprene, monoterpenes and sesquiterpenes, include double carbon bonds and therefore also react with O3:"

Page 4, line 78: ki is the bimolecular rate coefficient, not the pseudo-first-order rate coefficient.

Corrected.

Page 4, line 85: Note that measurements of HO2 reactivity have also been reported (Miyazaki et al., Rev. Sci. Instr., 84, 7, doi.10.1063/1.4812634).

The reference has been added.

Page 4, line 88: Are there other considerations for long-lived species? Is it necessary to assume that O3 is in steady state?

No, unless the other long-lived species also react with O3, in which case they contribute to the total reactivity in both the ambient atmosphere and the measurement. Equation 4 does not assume that O3 is in steady-state.

Page 4, line 97: Consider changing 'when photolysis is zero' to 'when photolysis rates are zero'.

Done.

Page 4, line 99: The comparisons between RO3 resulting from NO2, alpha-pinene and limonene are a little confusing. If NO2 has the lower rate coefficient it should require a greater concentration to reach the same O3 reactivity as alpha-pinene or limonene. For the rate coefficients given in Table 1, and assuming T = 298 K, p = 1 atm, 1 ppb of NO3 has RO3 = 8.7e-7 s-1, 2.7 ppb of alpha-pinene has RO3 = 6.4e-6 s-1 and 6.2 ppb of limonene has RO3 = 3.4e-5 s-1. Should this read that 2.7 ppb of NO2 has the same RO3 as 1 ppb of alpha-pinene and 6.2 ppb of NO2 has the same RO3 as 1 ppb of limonene? Please clarify.

We thank the reviewer for spotting this error which has been corrected. The sentence has been changed to: "1 ppb of NO2 has the same O3 reactivity as 0.37 ppb of a-pinene and 0.16 ppb of limonene (at 298 K)"

Page 5, line 127: The previous study describing measurements of RO3 defines it as the total O3 reactivity, and in their experiments/measurements assume [NO] = 0 and all observed RO3 is a result of reactions with VOCs. In this case, where NO is present and its effects on the observed RO3 has to be subtracted to give the O3-VOC reactivity, would it be sensible to define this as a separate parameter to RO3 where the O3-NO reactivity is known? This would avoid any future confusion between studies that may define RO3 as the total observed reactivity (as in the previous work) or as the subset of RO3 owing to O3-VOC reactivity (as in the current work).

The referee raises an issue which we considered carefully in adopting the definitions of ozone reactivity for this paper. The definition of R(O3) used here is consistent with the definition used in previous studies (Matsumoto, 2014 & 2016). We define R(O3) as the ozone reactivity due to BVOCs, and R(NO) as the ozone reactivity due to NO (see Eq. 6 on page 5). In the case of the previous studies, which were conducted in zero air, R(NO) = zero and the values of R(O3) are therefore comparable.

The following sentence has been added to Line 129: "This definition of R(O3) is directly comparable to the definition by Matsumoto (2014, 2016), since those experiments were conducted in zero air (i.e., at R(NO) = 0)."

Page 6, line 149: Please quantify the statement 'not substantially different'.

The difference between the model results is less than 0.1%. The number has been added to the text.

Page 7, line 169: Spelling of 'independent'.

Corrected.

Page 8, line 210: Please provide some further details regarding the requirements for the residence time. What difference in [O3] is required for accurate measurements of ozone reactivity? How much change in [VOC] is acceptable before the measurement of ozone reactivity is affected?

The required difference in O3 concentrations is not a fixed parameter, but depends on the BVOC loading. For the experiments described in this paper the drop in ozone concentration was typically of the order of a few ppb. We note however that the ratio between the two ozone measurements (before and after the reactor) is the more important quantity, as per Equation 5. The change in BVOCs must be such that pseudo first-order conditions are maintained for the duration of the residence time in the reactor. This is quantitatively discussed in Section 2.3 (lines 163-170).

Page 8, line 224: There is no hyphen in 'ad hoc' (also similar comments for in situ, 2 sigma, elsewhere).

Corrected.

Page 9, line 224: Please provide details of the mass transmission curve. What is the source? How does it affect the uncertainties in the measurements? What are the limits of detection for the VOC measurements?

We used the TO-14A aromatics standard mixture calibration gas and zero air to generate the mass transmission curve. The main uncertainty is that we are assuming that any compounds of interest that were not included in the mass transmission curve obey the same reaction kinetics in the reaction tube as the calibrated compounds. Holzinger et al. (2019) reported that a PTR-MS operated under standard conditions is able to accurately measure the concentration of uncalibrated compounds (to within 30%) if these compounds have high proton affinity and do not undergo unknown fragmentation using a mass transmission curve. This assumption is likely valid for BVOCs such as isoprene and monoterpenes (a-pinene and 3-carene), which were the primary focus during the laboratory plant experiments. Therefore based on the work by Holzinger et al., (2019), we are reasonably confident that we could quantify BVOCs levels, as we operated the PTR-MS using standard conditions (E/N of 129 Td). The reaction rate of many monoterpenes with H3O+ are well-known and we used the reaction rate for a-pinene (2.04e9 cm3 s-1 molecule-1) as this was expected to be a major constituent for the lemonthyme bag experiments. For the compounds that were included in the calibration gas, the limits of detection were calculated to be between 20-80 ppt, within the expected range (Sulzer et al. 2014). We assume that for BVOCs it is similar.

The clarify the calibration procedure using the mass transmission curve, we have amended the text between lines 231 and 236 as follows:

"The instrument (Sulzer et al., 2014) was operated according to the standard operating conditions recommended by the manufacturer (drift pressure = 3.8 mbar, drift tube temperature = 80 C and E/N = 129 Td), using $H_3O^+$ as the reagent ion. Calibration was performed using a TO-14A aromatics standard mixture (Airgas Inc., USA). This mixture does not contain biogenic compounds, so a mass transmission curve calculated using the calibration gas was used for quantification. Recent work by Holzinger et al. (2019) showed that a PTR-MS operated under standard conditions is able to accurately measure concentration of uncalibrated compounds (to within 30%) using a mass transmission curve, if these compounds have high proton affinity and do not undergo unknown fragmentation. This assumption is likely valid for BVOCs such as isoprene and monoterpenes (Holzinger et al., 2019), which we were the primary focus of this work. The limits of detection determined using zero air for calibrated compounds were in the order of 20-80 ppt."

Page 9, line 259: What is 'easy' about detecting the peak at m/z 59? Mass separation from other peaks? Peak height/ionisation cross-section for acetone compared to other compounds?

We chose acetone for these experiments as it is easily ionised by $H_3O^+$ ion and does not undergo fragmentation. Therefore it is straightforward to detect it by PTR-MS. To clarify the above, we have changed the text to read (line 258):

"Acetone was used as a tracer for these experiments because it is easily ionised by $H_3O^+$ and does not undergo fragmentation, and consequently is straightforward to detect by PTR-MS at its protonated mass (m/z 59)"

Page 10, line 261 onwards: What is the flow regime in the instrument? Is the assumption of plug flow appropriate? How was the concentration of NO determined in method 2?

We have assumed a plug flow inside the reactor (lines 268-269) only for the calculation with Method 3. The other two methods do not make this assumption, but implicitly take into account the flow regime in the reactor. The NO concentration was known (with an uncertainty of 5%), because we used a certified cylinder for those experiments. Line 265 has been modified as follows:

"Method 2 uses the TORS technique to measure the reactivity of $O_3$ (~20 ppb) with NO: the sample flow contained only ~100 ppb of NO (diluted from a certified gas cylinder, 4.90 +/- 0.25 ppm in $N_2$, by BOC UK) and, since the rate coefficient of NO + $O_3$ is known (1.89e-14 $cm^3$ $molecule^{-1}$ $s^{-1}$ at 298 K, with an uncertainty of 17% (Atkinson et al., 2004), the only unknown variable in Eq. 5 was the reaction time t."

Page 10, line 292: Please quantify the 'small but noticeable dependence of Rwall on humidity'.

The sentence has been changed to: "Rwall showed a weak dependence on the relative humidity inside the reactor (Rwall = 9.6e-7* RH + 4.4e-5, with R2 = 0.198)"

On line 298 "no observable pattern" was changed to "no clear pattern".

Page 11, line 295: State the temperature range in the text.

Added.

Page 11, line 298 onwards: What is the standard deviation and median of the measured ozone wall loss? Can the limit of detection be quantified more precisely using the observed variability in the ozone wall loss?

The median and the average standard deviation of the ozone wall loss have been added to the text. The standard deviation is variable within the same experiment and from experiment

to experiment (see Figure 5), which means that the detection limit is also variable (Equation 6). The average standard deviation of the ozone wall loss (2.4e-5 s-1) corresponds to a reactivity of ~10 ppb of a-pinene. As indicated in Section 5.1 (lines 350-351), below 10 ppb the measured reactivities are statistically indistinguishable, but since the wall loss can be higher than that, the estimated detection limit can also be larger. The following changes were made to the manuscript:

Lines 300-301: "The average standard deviation of R(wall) was 2.4e-5 s-1 and the interquartile range was 0.5-1.2e-4 s-1 (mean = 9.8e-5 s-1 , median = 7.1e-5 s-1 ), which corresponds…"

Page 11, line 311: Please provide further details of the experiments that led to '. . . eventually settling on a sample flow of ~2.3 slpm'. What were the ranges of conditions investigated? How did the instrument perform under these conditions? Why was a flow of ~2.3 slpm considered optimum?

We experimented with several values of the ozone flow between 500 and 2500 sccm. The sample flow was adjusted accordingly, so that all the flows of the system were balanced. We settled on a combinations of flows which worked well with the conditions we had in the laboratory experiments, but we recognize that not all the possible combinations of flows have been explored. Therefore it is not possible to say that these are the absolute optimum settings for the system. This is indeed one of the objectives of the future work on the instrument, as indicated in the Conclusions.

Page 11, line 323: What is the impact of the difference of ~1 ppb on the uncertainty in the ozone reactivity measurements?

There is no impact. As mentioned on lines 321-322, we apply a correction factor to the ozone data to account for the difference between the monitors.

Page 12, line 345: How well did the concentrations determined from the diffusion tubes agree with those determined by the PTR-MS measurements?

The agreement was, on average, within 14%. The text has been modified as follows:

"The concentration of a-pinene was then calculated from the diffusion rate and confirmed via direct measurements by PTR-MS (Sect. 3.2) with an agreement of ~14%."

Page 12, line 348: Please quantify 'reasonable agreement'.

The sentence has been changed to "The agreement between the calculated and measured ozone reactivity for a-pinene mixing ratios larger than 40 ppb was about 25% -- within the combined uncertainties of the instrument and of the a-pinene + O3 rate coefficient (41%, Atkinson et al. (2004)) "

Page 12, line 358: The range of values for the limit of detection are 1/3 to 2/3 of that described previously, 'comparable' is somewhat subjective. How does the residence time affect this? A more detailed description of the instrument used in previous studies would be helpful to provide the reader with a more informed comparison.

A detailed description of the instrument used in previous studies is beyond the scope of this paper, but can be found in Matsumoto (2014). The observed ozone change is inversely proportional to the residence time (see Equation 5), and therefore the instrument detection limit also is inversely proportional to the residence time (subject to the limitations in the consumption of BVOCs in order to maintain pseudo first-order conditions). Therefore, if the residence times of the two instruments differ by a factor of about 2, the corresponding detection limits can be expected to vary by roughly the same amount.

Page 13, lines 372-375: What were the sources of the Teflon bag, halogen lamp and small fan?

Added.

Page 13, line 376: What is meant by 'the natural humidity of the plants'? Natural release of water vapour via transpiration and evaporation by the plants?

That is correct. The sentence has been modified as suggested.

Page 13, line 383: What are the uncertainties in the stated values?

Given the variability of the data (Figure 7) it makes more sense to state the interquartile range, because the standard deviation of the timeseries would be of the same order of magnitude of the mean. We refer to Section 5.1 for a discussion of the instrumental uncertainties (see above).

Page 14, line 408: Please provide some approximate quantification for the statement 'concentrations of BVOCs . . . are higher and the concentrations of NO lower . . .'. Page 15, line 458: What is the basis for the assumption of NO in the reactor being ~20% of the ambient concentration?

Given the location of the experiment– a suburban area in central England – we feel that it is reasonable to assume that the levels of BVOCs inside the glasshouse were higher than in the nearby environment. Unfortunately, we cannot be quantitative in the absence of simultaneous measurements of BVOCs inside and outside the glasshouse.

As for the issue of NO levels inside the glasshouse and related discussion we have decided to remove this section and Figure 9 from the paper. Please see the extended discussion in our reply to reviewer #2.

Page 14, line 415: What was the variability in the measured wall loss?

The following was added to the text: "The ozone wall loss during the measurement period varied between 4.9e-5 and 1.1 e-4 s-1 (first and third quartiles), with mean values between 0.7e-4 and 1.1e-4 s-1."

Page 14, line 420: What were the most important species?

This information is on the following page (lines 432-433).

Page 14, lines 422 and 423: What are the uncertainties in the stated means?

Given the variability of the data (Figure 8) it makes more sense to state the interquartile range, because the standard deviation of the timeseries would be of the same order of magnitude of the mean. We refer to Section 5.1 for a discussion of the instrumental uncertainties (see above).

Page 15, line 449: Change '. . ozone reactivity tends peak . .' to '. . ozone reactivity tends to peak . .'

Done.

Page 16, line 476: It's not clear how the listed improvements will be achieved or how TORS will be able to make ambient measurements (line 479). Specific details would help to avoid this simply reading as a wishlist.

The paragraph is intended to highlight the areas of future development for the TORS technique. It also provides guidelines for other researchers that may be considering using this approach. We have changed to text as follows:

Line 476: "Further work will improve" changed to "Further work will aim to improve"

Line 478-479: "In the future, TORS will be able" changed to "With these improvements and proper supporting measurements, the detection limit and the uncertainty of TORS can be improved and the technique will be able to make measurements under a wider range of conditions"

Also, for consistency, the following changes were made to the abstract and the conclusions:

Lines 6-7: changed to "larger than typical ambient levels, but observable in environmental chamber and enclosure experiments, as well as in BVOCs-rich environments"

Lines 470-472: changed to "These mixing ratios are larger than typical ambient levels, but can be observed in BVOCs-rich forested environments and in enclosure studies (Duhl et al., 2008; Bouvier-Brown et al., 2009, Kammer et al., 2018), and can easily be reproduced in laboratory and environmental chamber experiments."

---

## Author Comment (AC2) · 7 Jan 2020

**Response to Anonymous Referee #2**

A. General comments:

In this manuscript, authors focused on total ozone reactivity, built and characterized their instrument in the laboratory, and tested it in the glasshouse. This study is positioned as a basic research of the instrument and a demonstration for measuring BVOCs emission from plants. Ozone reactivity of BVOCs is interesting for investigating BVOCs. Thus, the reviewer believes that this work has an important implication and is significant enough to be published in this journal. However, the present manuscript leaves several points to be improved, clarified, modified, and/or reconstructed, in order for readers to understand descriptions and to recognize the significance of this study clearly. Especially, it is necessary to correct critical errors on quantitative descriptions, to indicate more information and explanations, and to clarify the story of discussion which authors want to express during the demonstration of the instrumental test.

We thank the referee for the detailed comments and suggestions. Below are our responses and related modifications to the manuscript. The line numbers refer to the version of the manuscript published on AMTD.

B. Important specific comments:

B1) Line 101 and followings: Quantitative descriptions on contribution of NO2 should be corrected. The reviewer thinks that 1 ppbv of NO2 corresponds to 3.52/9.6 = 1/2.7 = 0.37 ppbv of a-pinene and 3.52/22 = 1/6.25 = 0.16 ppbv of limonene, respectively. And then, consequently, based on the correct values, descriptions on importance of nighttime NO2 should be checked again, including descriptions in Sect. 5. Please do not mislead readers.

We thank the reviewer for spotting this error which has been corrected. The point, however, stands: the formation of NO3 (and hence the loss of O3 to NO2) is significant only when NO2 concentration is high, which is largely not the case in the conditions TORS was designed for.

Lines 102-103 were modified to: "This means that NO2 can be a significant ozone loss during the night only when its concentration is high, which is not usually the case in unpolluted forested environments."

For the impact of NO2 on the glasshouse measurements presented in Section 5, see below.

B2) Fig.1 and Lines 160 ~ (descriptions on model estimations) : Please show more information and descriptions on model estimations. For example, how much is the initial OH concentration? Why does the delta-OH in Fig.1 distribute less than zero? At all, what is 'delta-OH' in Fig.1? (No descriptions and explanations in sentences.) Additionally, please add descriptions on the applied reaction time in the sentence and in the caption of Fig.1.

We recognize that Figure 1c (and related text) could be clearer, so we have changed it to show instead the mean OH concentration at each concentration of a-pinene as a function of initial cyclohexane. The corresponding paragraph in Section 2.3 (lines 170-177) was modified as follows:

"The average modelled concentrations of the OH radical at different levels of cyclohexane are shown in Fig. 1c. In the absence of cyclohexane, the model calculated OH concentrations between 1.3e6 and 4e6 molecule cm-3 for a-pinene mixing ratios of 0.1 ppb and 50 ppb, respectively. With addition of the OH scrubber, the simulated concentration of OH in the reactor decreased by 2 orders of magnitude at mixing ratios of cyclohexane between 1 ppm

(for a-pinene <5 ppb) and 10 ppm (for a-pinene = 50 ppb). Increasing the cyclohexane mixing ratio above 10 ppm did not cause further decrease in the calculated concentration of OH, nor a reduction in the loss of ozone and a-pinene (Fig. 1a-c), at least within the range of a-pinene concentrations explored by the model."

B3) Line 204: Why and how can 'ambient temperature and pressure' affect the chemistry inside the reactor? Now there are no explanations in the manuscript.

Temperature and pressure affect the rate coefficients of the reactions that take place inside the reactor. The sentence has been corrected to: "and, to a lesser extent, ambient temperature and pressure (which influence the rate coefficients of chemical reactions). To date, the effect of temperature and pressure is negligible, as the system has been operated under near-ambient conditions."

B4) Line 258 'simultaneously' : How did authors check and ensure the simultaneity? Uncertainties on the timing of the synchronized injection of acetone can cause uncertainties on the determined reaction time.

The injections were made manually by two different persons using a timer for synchronization. The reviewer is correct that this procedure has inherent uncertainties, however the time difference is such that these are very small. For a residence time of 164 seconds (Figure 3) a discrepancy of, for example, 0.5 seconds in the injection times results in an error of ~0.3%. The other approaches also have their own uncertainties and assumptions, which is the reason we have used three different methods to determine the residence time.

B5) Fig.4 & Sect.4.1 (especially, on NO+O3 reaction): In 'Method 2 (Fig.4a)', k[NO] up to 0.05 (maybe in the unit 's-1'? please clarify the unit;) were adopted. It means that [NO] up to about 100 or 110 ppbv, I think. I want to know authors' opinions on following questions and for authors to revise the descriptions to clarify the situations:

[Q1] Concentrations of NO and O3 are similar. The reaction NO+O3 is fast. Then, both NO and O3 can significantly decrease within the reaction time. (Question:) Are the settings of the experiments, calculations, and discussion to determine the reaction time proper to realize and ensure the 'pseudo first-order reaction' as described in Line 267 ? When I tried a rough and simple calculation on the decrease of O3 and NO, for example, till t = 150 sec, NO decreased from 100 ppbv to 5 ppbv and O3 decreased from 120 ppbv to 25 ppbv. NO can drastically decrease during the reaction time due to its fast reaction with O3.

The experiments with NO were conducted at lower concentrations of O3 (~20 ppb) than the normal operating conditions of TORS. During the reaction time the concentration of NO therefore dropped only to 80-90 ppb. We do not think that the fact that these experiments were conducted at lower ozone levels than the other experiments is important, since they were only used to determine the residence time. The NO and O3 concentrations used for these residence time experiments have been added to the text.

In any case, the reviewer is correct that the system deviates from pseudo first-order conditions during these experiments. We have reanalyzed the data with the help of a simple steady-state model. As shown in the Figure below, because the reaction between NO and O3 is very fast, the concentration of NO decreases to 80 ppb within 60 seconds and, therefore, the initial concentration of NO is not representative of the NO concentration inside the reactor for most of the residence time.

[Figure]

We have therefore recalculated the residence time with method 2 using the average NO concentration over a period of 150 seconds, instead of the initial NO concentration. The results of this reanalysis are now shown in Figure 4 (the calculation using the initial NO concentration is also shown for just one flow, for reference). The residence time obtained in this way is closer to the residence time calculated assuming a plug flow (method 1) than the one obtained using the initial NO concentration. To make the analysis more robust, a best fit polynomial function (2nd degree) was calculated using all 3 methods and the results of the fit (140 seconds) has been used as the residence time for all the experiments in the paper. The new residence time is 9% higher than the one used in the original version of the paper, which is well within the estimated total uncertainty of TORS (32%). All ozone reactivities and ozone wall losses mentioned in the paper have therefore been recalculated (they are 9% lower) and the relevant figures corrected accordingly. Additionally, the following modifications were made to the text:

Line 267: added "Since the chemical system deviates from pseudo first-order conditions, the mean concentration of NO inside the reactor was used to analyze the experimental results."

Lines 274-275: changed to "In the experiments described in Sect. 5, we used a value of 140 seconds for the residence time, determined by fitting a 2nd degree polynomial to all three methods, as shown in Fig. 4b for a reactor flow of 2470 sccm."

The caption of Figure 4 was changed to "(a) NO reactivity experiments analyzed using the mean NO concentration inside the reactor. One experiment analyzed using the initial NO concentration is also shown, for reference. (b) Residence times as a function of reactor flow – determined by three different methods – and polynomial fit to the three methods (black dashed line). The results of the acetone injection method are taken from Fig. 3. The red star indicates the residence time used in this work (140 seconds for a reactor flow of 2470 sccm)."

[Q2] In Fig.4a, we can estimate the reaction time. For example, in case of 2470 sccm, for k[NO] = 0.04 (s-1?), -LN(O3(t)/O3(0)) is about 4, and the slope of the regression line is roughly found as 4/0.04 ~ 100 sec. However, for 2470 sccm in Fig.4b, we can read out the reaction time determined by NO reactivity as about 130 sec. Which figures are correct? And is the reaction time that authors determined and described in the manuscript exactly correct? These are critical points because the reaction time is one of the most important factors to determine ozone reactivity. Would you please confirm them and, as necessary, revise descriptions in order to clarify authors' standpoints?

Figure 4 has been substituted with a new version following the comments by the referee on the NO+O3 experiments to determine the residence time (see above). Note that as a consequence of the modifications listed above, the residence time was changed throughout the paper to 140 sec (9% higher than the original value of 128 sec).

Additionally, as an associated question, what value of k(NO+O3) was applied in this study? Authors indicate the values as '1.89x10- 14' in Table 1 and '1.73x10-14' in Line 266. The ratio 1.89/1.73 is 1.09. If authors mistake the values of k, it results in ~ 10 % errors by itself.

The value used in this work is 1.89e-14, as recommended in the updated datasheet from Atkinson et al. (2004). The text on line 266 has been corrected.

B6) Fig.6a and Sect. 5.1.: Between calculations and experiments, the dependence on a-pinene concentrations can be seen different. Experimental data rise up steeply (a-pinene < 10 ppb) and then gradually increase (with smaller slope than calculations). Calculations show a straight line. Would you please explain briefly these results in the sentences? What happens in the reactor, do you think?

The ozone reactivity at a-pinene levels below 10 ppb is less than 2.3e-5 s-1, which is lower than the interquartile range of the wall loss (see line 300), and corresponds to a drop in the O3 mixing ratio inside the reactor of <0.4 ppb, which is less than the detection limit of the ozone monitors. Therefore we think that those data points are effectively indistinguishable from the instrument's noise.

The sentence at line 350 has been modified as follows: "At mixing ratios below 10 ppb of a-pinene, the measured reactivities cannot be statistically distinguished from each other and from zero; in fact, the corresponding reactivity (2.36e-5 s-1) is of the same magnitude as the average standard deviation of R(wall) (Sect. 4.2)".

B7) Sect. 5.3, about NO contribution to ozone reactivity: It is unreasonable to understand that the present descriptions that NO concentration in the house is assumed as 20 % of ambient concentration (7 km far from the experimental site). We think that the assumption on NO concentration is conveniently and arbitrarily made. If authors have any information to validate it, it is necessary to show the evidence and to explain them clearly. If not, it is necessary to indicate clearly authors' opinion, procedures of analysis, flow of argument, and the positioning of this experiment. Especially, please distinguish between the fact and the interpretation. For example, the reviewer's understanding of this section is as follows . . . (Fact) This experiment is aimed at a challenging demonstration of the TORS instrument. The major point is ozone reactivity measurement. It is true that significant RO3 was captured during the experiment. (Assumption & Caution) Authors want to discuss on BVOCs contribution to captured RO3 data. To do it, NO contribution should be considered. However, NO concentration in the sample is not monitored. Then the monitoring data (7km far) were adopted in order to know rough information on NO concentration. However, NO concentration can vary largely in the urban area due to the location, traffics, time of day, etc. Thus, adopted NO concentration is arbitrary and has large uncertainties. (It is unfair if descriptions on the limitations are insufficient.) (Interpretation & Authors' opinion) (For example) To discuss NO contribution, NO concentration was assumed as 0 to 100 % of ambient (7km far) concentration. When NO was set to ?? % or more, ozone reactivity by NO is larger than captured RO3. So it was suggested that NO concentration was less than ?? % of ambient level. In this study, as an upper limit (?), NO was assumed as 20 % of ambient level where NO contribution is equal to and/or less than captured RO3 (Fig.9). Then BVOCs contribution to RO3 was indicated as ?? in Fig.9.

B7-2) Line 441 'urban background site' and Line 457 '50m from nearby roads': Please clarify the positioning of each site. Is the background site far from roads (i.e. not 'roadside')? Is the experimental site also considered as 'urban background'?

B7-3) Fig.8: NO concentration data at the urban background site (or NO contribution to RO3) should be indicated, because they are essential for us to discuss NO contribution to RO3 in Fig.9. For example, does NO really show the daytime peak as Fig.9? Does NO indicate its variation similar to RO3? B10) Around Line 455, about the lifetime of NO by O3 reaction: k is about 2 x 10ˆ-14, [O3] (30 ppb) is about 7.5x10ˆ11, then k[O3] is about 1.5x10ˆ-2 s-1. Thus, the lifetime of NO is about 67 sec ( = 1/(1.5x10ˆ-2 s-1) ). This error is critical for the authors' consideration that NO can be reduced to 20 % of ambient level due to O3 reaction in the sample tube (4 sec). Associated to the comment B8, can ambient NO react with residual O3 in the glasshouse before the sampling inlet? Please consider them again and reconstruct the descriptions.

B7-4) Such authors' standpoint and assumptions should be noted at the beginning of the paragraph, as well as at the end. 'This is only a (challenging?) rough estimation & interpretation based on some assumptions', 'NO concentration is not monitored; NO contribution is assumed', for example. Also in the caption of Fig.9, such descriptions on 'assumption' are desirable.

B8) Sect. 5.3, about O3 and NO2 in the glasshouse: Authors consider that NO concentration in the house is smaller than ambient. Then, how about O3 and NO2 concentrations in the house? Is the glasshouse enclosed by walls? Or, can ambient air pass through the house? As a result, how much are O3 and NO2 concentrations in the house, do you think? Are O3 and NO2 in the house significant for RO3 measurement?

The reviewer is correct that the assumptions on NOx levels inside and outside the glasshouse, and the usage of monitoring data collected far from the observations (although under similar conditions), are speculative. Reviewer #1 also made similar comments.

In the absence of NOx measurements inside the glasshouse, this part of the paper was meant to be only qualitative, but we realize that we don't have enough information to support the assumptions made. The important point here is that the ozone reactivity measurements showed in Figure 8 do include a contribution from ambient NO, although we cannot quantify it. This implies that accurate NO measurements must always be taken alongside ambient ozone reactivity measurements.

Therefore, we have removed Figure 9 and lines 452-463 from the final version of the manuscript and changed lines 438-442 to: "Measurements of NO were not available inside the glasshouse, so it is not possible to quantify the contribution of R(NO) to the total ozone reactivity measurements shown in Fig. 8."

The caption of Figure 8 was changed to specify that the figure shows the sum of R(O3) and R(NO).

We have also added the following sentence to the conclusions (Section 6): "Moreover, our experimental data indicate that accurate measurements of NOx are always required to be able to interpret the TORS observations."

B9) Fig.9, about the diurnal variation of BVOCs contribution to RO3: Why does the BVOCs contribution indicate their peak during night (or before dawn)? Because BVOCs emission from plants usually depend on temperature and light intensity, it is expected that BVOCs

contribution to RO3 has daytime peak. Would you please add explanations and your opinions on such diurnal variation of BVOCs contribution?

This is mentioned on lines 441-451. BVOCs emissions are higher during the day but OH concentrations are also higher during the day. Most BVOCs react more readily with OH than with O3 and, as a consequence, the contribution of BVOCs to total RO3 is reduced during the day.

C. Other comments and Technical corrections:

C1) Line 29: 'state' —> states' ?

Corrected.

C2) Lines 64 - 65 and References: If authors want to refer the proceeding of a conference (Park et al.(2013), another earlier proceeding in the previous conference should be referred: Matsumoto, J.: Comprehensive analyzer for biogenic volatile organic compounds detected as total ozone reactivity, in AGU 2011 Fall Meeting, 2011.

The reference has been added.

C3) Line 71 'known BVOC mixture' —> The reviewer cannot find descriptions on such 'mixture' in the following sections. Please clarify what the mixture is. (e.g. mixture of a-pinene and cyclohexane? but the scavenger is not BVOC . . . )

The sentence has been modified to: "laboratory measurements with known concentrations of selected BVOCs,"

C4) Line 90 (eq.2) and Line 105 (eq.3): Ozone concentrations, [O3], are missing in all loss terms of ozone.

Corrected.

C5) Line 91 'R11' —> 'R2', too? C6) Line 131 'R5-R8' —> 'R6-R8'?

The hyphen indicates a range, so "R1-R11" includes R2, and "R5-R8" includes R6.

C7) Line 140 'Sect.4.2 —> Sect.5.1, too?

The reference is to the experiments done to characterize the ozone wall loss, which are described in Section 4.2. Section 5.1 does not discuss the ozone wall loss, only the experiments with BVOCs.

C8) Line 146: Please add brief descriptions on 'Sommariva et al., submitted'.

The full reference has been added.

C9) Lines 178-184 and Fig.1d: In Fig.1d, RO3(with)/RO3(w/o) (please add the axis name in the figure) are ~ 0.965 at a-pinene = 0.1 ppb and ~0.950 at 0.5 ppb. That is, RO3(w)/RO3(wo) ratio decreases while a-pinene < 0.5 ppb, and then increase for a-pinene > 0.5 ppb. Would you please add brief explanations on this trend?

It is difficult to see because of the scale of the figure, but the point at 0.95 corresponds to 1 ppb of a-pinene not 0.5 ppb (the point at 0.5 ppb was actually missing from the figure because of a plotting error, which has now been fixed).

The values in the Figure are as follow:

| Initial a-pinene mixing ratio (ppb) | ratio of R(O3) at 1ppm of cyclohexane to R(O3) without cyclohexane |
|:---:|:---:|
| 0.1 | 0.96 |
| 0.5 | 0.94 |
| 1 | 0.95 |
| 5 | 0.98 |
| 10 | 0.99 |
| 50 | 1.01 |

In other words, for levels of a-pinene below 1 ppb, the effect of the OH scrubber is inversely correlated to the concentration of a-pinene, while above 1 ppb it is directly correlated (as the reviewer correctly points out). This is most likely due to the balance between OH production by a-pinene ozonolysis vs OH consumption by a-pinene and/or cyclohexane, which is different at different levels of a-pinene.

C10) Line 193 'ambient measurements' —> Strictly, 'ambient, not always, but nighttime & high NOx & low NO' ? Please clarify the conditions.

The reviewer is correct that high NOx conditions are required, although we note that nocturnal conditions are not required, since the sentence refers to the reactions taking place inside the reactor, which is dark.

The sentence has been modified to: "NO3 formation in the reactor is only an issue for ambient measurements under moderate or high NOx conditions, not for laboratory, enclosure and environmental chamber experiments under low or zero NOx conditions."

C11) Line 264 'about 30 %': It is true that '164 s' is 70 % of '228 s' and thus 30 % smaller than '228 s'. However, '228 s' is 139 % of '164 s' and thus about 40 % larger than '164 s'. The descriptions are not clear, including the word 'difference'.

The sentence has been changed to: "the residence times estimated using the "means" calculations are about 40% larger than those estimated using the "maxima" calculations"

C12) Line 267 'Eq.3' is not correct.

Corrected to "Eq. 5".

C13) Around Line 294: Please clarify, anywhere in the sentences earlier, what the 'central? 0.5 slpm flow of zero air (without O3 lamp and scrubber)' in Fig.2 means. Then the mean of 'dilution' (Line 293) can be clear. (The zero flow may be used as 'dilution flow to control the concentration of ozone' as described in Line 312. However, before Line 294, we have not already recognized that point.)

We have amended Figure 2 to clarify the components of the system. Line 216 has been changed to: "An ozone flow is generated by irradiating a flow of zero air with a UV mercury lamp (UVP Ltd., UK); a zero air flow is added downstream the mercury lamp to control the concentration of ozone (Fig. 2)."

C14) Fig.8 'estimated HIGH & LOW': Why do these data indicate diurnal variation? How to determine these data?

The oscillations in the estimated reactivities are due to ambient temperature, which affects the calculation of the rate coefficients. The description of how these estimates were derived is on page 15 (Section 5.3).

C15) Scientific names of plants: Italics?

The scientific names of all plants have been written in italic.

C16) Line 470: It is desirable when conditions like averaging time and reaction time are also indicated.

Added on line 469: "with a residence time of 140 seconds, an averaging time of 5 minutes, and an estimated total uncertainty of ~32%"

C17) Table 1: If possible, please indicate the lifetimes of VOCs for O3 = 120 ppbv. Then we can compare the lifetimes to reaction time and discuss the pseudo first-order reaction.

The table has been modified as suggested.

C18) Figures: Please add the names (titles) of axis.

Done.